



# Investigating controls of shell growth features in a foundation bivalve species: seasonal trends and decadal changes in the California mussel

Veronica Padilla Vriesman[1], Sandra J. Carlson[1], Tessa M. Hill[1]

[1]Department of Earth and Planetary Sciences, University of California, Davis, One Shields Avenue, Davis, CA, 95616, USA

*Correspondence to*: Veronica Padilla Vriesman (vpvriesman@ucdavis.edu)



**Abstract.** Marine bivalve mollusc shells can offer valuable insights into past oceanographic variability and seasonality.

Given its ecological and archaeological significance, *Mytilus californianus* (California mussel) presents the opportunity to examine seasonal and decadal changes recorded in its shell over centuries to millennia. While dark–light growth bands in *M. californianus* shells could be advantageous for reconstructing past environments, uncertainties remain regarding shell structure, environmental controls of dark–light band formation, and the amount of time represented by a dark–light pair. By analyzing a suite of *M. californianus* shells collected in 2002, 2003, 2019, and 2020 from Bodega Bay, California, we

describe the mineralogical composition, establish relationships among growth band pattern, micro-environment, and collection season, and compare shell structure and growth band expression between the archival (2002–2003) and modern (2019–2020) shells. We identified three mineralogical layers in *M. californianus*: an outer prismatic calcite layer, a middle aragonite layer, and a secondary inner prismatic calcite layer, which makes *M. californianus* the only *Mytilus* species to precipitate a secondary calcite layer. Within the inner calcite layer, light bands are strongly correlated with winter collection

months and could be used to reconstruct periods with moderate, stable temperatures and minimal upwelling. Additionally, modern shells have significantly thinner inner calcite layers and more poorly expressed growth bands than the archival shells, although we also show that growth band contrast is strongly influenced by micro–environment. *Mytilus californianus* from northern California is calcifying differently, and apparently more slowly, than it was 20 years ago.

## 1 Introduction

Marine bivalve shells offer a complex yet valuable record to explore questions about paleo– and modern seasonal extremes since many bivalve species can record ambient conditions as they calcify (Jones and Quitmyer, 1996; Wanamaker Jr. et al., 2006; Welsh et al., 2011; Schöne and Gillikin, 2013; Trofimova et al., 2021). Detailed paleo–environmental, paleoceanographic, and anthropological information can be extracted from shell growth records over multiple time scales depending on the species' shell growth rate, the accuracy of ontogenetic age estimates, and the periodicity of the growth

band pattern within the shell (Hallmann et al., 2013; Jazwa and Kennett, 2016; Cannon and Burchell, 2017). Seasonally resolved shell growth features allow for the approximation of historic baselines of temperature variability, the comparison of inferred paleo–temperature extremes to modern temperature ranges, and the prediction of climatic change impacts on organisms and ecosystems, which are typically controlled more strongly by extremes (Sydeman et al., 2014; Poloczanska et al., 2016; Mellin et al., 2016) than they are by average conditions. During calcification, both environmental and biological

factors influence shell characteristics and chemistry differently depending on the species (Table 1). In order to interpret the shell growth features of a particular species as a paleo–seasonal or paleoceanographic archive, a clear understanding of the environmental parameters influencing the appearance of the growth band pattern and the timing of shell growth is required. For example, the well–studied bivalve species *Arctica islandica* has served as a reliable climate record with sub–seasonal resolution for high–latitude marine environments because it has been determined to be long–lived (~ 500 or more years) with

clearly expressed annual and daily growth lines that form continually throughout the year (Schöne et al., 2005a; Schöne,



2013). One individual provides high–resolution environmental information and seasonal extremes for multiple centuries within a single shell (Weidman et al., 1994; Schöne et al., 2005a; Schöne, 2013). While *A. islandica* is a unique archive due to its longevity and regularity, the relationships between environmental parameters and shell growth features should be tested in other bivalve species to optimize the paleoceanographic utility of fossil, archaeological, and historic specimens.

**Table 1**. Chemical and microstructural growth features recorded in bivalve shells and their environmental and biological influences. Species examples are not exhaustive.

| Shell feature | Environmental influences | Biological influences | Species examples |
|---|---|---|---|
| Light banding | Representative of warm temperatures (summer), high tide, high food availability, or conditions that allow for normal growth | Aerobic metabolism, faster calcification rate | *Mercenaria mercenaria* (Lutz and Rhoads, 1977); *Crassostrea virginica* (Surge et al., 2001) |
| Dark banding | Representative of cold temperatures (winter), low tide, low food availability, or conditions that impede normal growth | Anaerobic metabolism; slower calcification rate | *Mercenaria mercenaria* (Lutz and Rhoads, 1977); *Crassostrea virginica* (Surge et al., 2001) |
| $\delta^{18}O_{[shell]}$ | Inversely correlated with seawater temperature and positively correlated with salinity | Growth slowdown or shutdown prevents the shell from recording the full annual range of $\delta^{18}O$–inferred SST | *M. californianus* (Ford et al., 2010); *M. galloprovincialis* (Zhao et al., 2019*); Pecten maximus* (Freitas at al., 2012); *Saxidomus gigantea* (Hallmann et al. 2011) |
| $\delta^{13}C_{[shell]}$ | Inversely correlated with upwelling strength; upwelling delivers remineralized $^{12}C$ to surface waters | Metabolic carbon is incorporated into $\delta^{13}C_{[shell]}$ during respiration; photosymbiosis | *M. californianus* (Killingley and Berger, 1979; Pfister et al., 2011; Ferguson et al., 2013); *Tridacna* species (Killam et al., 2020) |
| $Mg/Ca_{[shell]}$ | Positively correlated with seawater temperature | Strong and positive relationship between Mg/Ca and growth rate | *M. californianus* (Ford et al., 2010); *M. trossulus* (Klein et al., 1996); *Pinna nobilis* (Freitas et al., 2005) |
| Ba/Ca | Inversely correlated with salinity; freshwater input proxy due to higher [Ba] in rivers relative to seawater | Potential remobilization of Ba stored in tissue during spawning | *M. edulis* (Gillikin et al., 2006); *Ruditapes philippinarum* (Poulain et al., 2015) |





### 1.1 *Mytilus californianus* as a study organism

One species of interest for the northeastern Pacific Coast is *Mytilus californianus* (California mussel), an ecologically and
culturally significant intertidal species that spans 20° of latitude ranging from the Aleutian Islands of Alaska to Baja
California in northern Mexico (Paine, 1974). *Mytilus californianus* is a foundation species in rocky intertidal environments,
playing a critical role in structuring and maintaining the intertidal community. It provides a habitat for ~ 300 interstitial
species, filters detritus from seawater during feeding, and serves as a food source for a variety of predators (Paine, 1974;
Smith et al., 2009; Connor et al., 2016). *Mytilus californianus* shells are abundant in coastal California shell middens that
span the terminal Pleistocene (~ 12,000 BP) through the late Holocene in age; many Indigenous communities, including the
Coast Miwok, Island Chumash, and Salinan peoples, harvested California mussels as a critical or even primary food source
(Jones and Richman, 1995; Jones and Kennett 1999; Kennedy, 2004; Kennedy et al., 2005; Braje et al., 2007; 2011; 2012;
Campbell and Braje, 2015). Shellfish collection remains an important element of Traditional Ecological Knowledge and is
still practiced by Indigenous communities along the West Coast of North America (Lepofsky et al., 2015). Given its
prevalence in the archaeological record and abundance in modern intertidal ecosystems, *M. californianus* provides the
opportunity to reconstruct coastal environments at various temporal scales and explore variation in shell growth features in a
single species over the past ~ 12,000 years from multiple northeastern Pacific Coast sites. However, our understanding of
shell structure and calcification rates and timing in *M. californianus* is highly variable and site-specific (e.g., Blanchette et al.
2007; Smith et al. 2009; Ford et al. 2010), hindering interpretations in fields ranging from archaeology to paleoceanography.

In order to accurately interpret *M. californianus* shell growth features – and to determine whether *M. californianus*
can serve as a reliable paleo–archive – we analyzed the shell morphology, mineralogical layering, and growth band pattern
of 40 specimens collected over various seasons in 2002, 2003, 2019, and 2020 in conjunction with sea surface temperature
(SST) records and upwelling indices for the same location (Bodega Bay, California, 38.3332° N, 123.0481° W). We aimed
to first characterize the shell structure and mineralogical layering of *M. californianus*, and then focused closely on the
growth band pattern in order to investigate environmental controls on shell growth to address the following questions: (1)
What is the influence of micro–environment (tidal position and habitat type) on the visual expression of growth bands in *M.
californianus* shells from Bodega Bay?  (2) What is the influence of oceanographic conditions (SST and upwelling intensity)
on the coloration (dark or light) of growth bands? (3) Are there temporal or periodic environmental trends (seasonal, annual,
decadal) influencing shell growth patterns (growth band expression and coloration of growth bands)?

Light bands in bivalve shells represent increments of normal growth while dark bands indicate slow or stunted
growth during periods of stressful or sub–optimal conditions (Lutz and Rhoads, 1977; Killam and Clapham, 2018). Dark
banding is possibly the product of increased organic material relative to calcium carbonate production during anaerobic
conditions, or the visual representation of changes to the crystal microfabric when calcification occurs more slowly. In either
case, however, dark bands are associated with reduced calcification. Using this proxy, we aim to link environmental



conditions with growth band coloration to determine the timing and periodicity of dark–light band formation in *M. californianus* populations from northern California, and whether there are spatial or temporal differences in growth patterns.

## 2 Methods

### 2.1 Oceanographic setting

Bodega Bay, California is located approximately 100 km north of San Francisco Bay in the central portion of the California
Current System (CCS). Oceanographically, Bodega Bay experiences strong seasonal cycles: a spring–early summer (March to July) upwelling season with low mean monthly SSTs (~ 10 to 12°C), a late summer–fall relaxation season (August to November) with reduced upwelling and relatively warmer monthly SSTs (~ 13 to 15°C), and a cool winter (December through February) with heavy precipitation and moderate SSTs (García–Reyes and Largier, 2010; 2012). The CCS comprises the dominant south–flowing California Current, the subsurface north–flowing California Undercurrent, and the
seasonal north–flowing Davidson Current present at the sea surface in winter (Hickey and Banas, 2003). The geometry of the California coastline interacts with the CCS to produce different temperature regimes in northern and southern California; north of Point Conception, the coastline is roughly parallel to alongshore winds, resulting in high Ekman transport and strong upwelling near the coast (Huyer, 1983; Checkley and Barth, 2009). Interannual and decadal regional variability within the CCS is largely driven by El Niño Southern Oscillation (ENSO) and Pacific Decadal Oscillation (PDO) (García–Reyes and
Largier, 2012). In addition to ENSO and PDO phases, local–scale coastal variability on the order of meters to a few kilometers is controlled by local surface warming and wind stress (Dever and Lentz, 1994; García–Reyes and Largier, 2012).

For this study, three intertidal collection locations were chosen: Horseshoe Cove (a protected marine environment in Bodega Marine Reserve, BMR), an open–coast BMR site (350 m north of Horseshoe Cove), and a third site at Portuguese Beach (an open–coast site 7 km north of BMR). All three collection locations are located along Sonoma Coast, west and
northwest of Bodega Harbor. Sonoma Coast is considered one oceanographic region since it is part of the same upwelling cell within the CCS (Largier et al., 1993; Wing et al., 1995), and 7 km of alongshore separation results in the same coastal SST and upwelling patterns at BMR and Portuguese Beach (J. Largier, pers. comm., 2021).

### 2.2 Specimen collection and preparation

To examine the environmental and temporal factors influencing growth band patterns, we analyzed shell growth features
from 40 *M. californianus* samples (n = 27 shells from 2019 and 2020; n = 13 shells collected in 2002 and 2003). Specimens were categorized as either modern (collected in 2019–2020) or archival (collected in 2002–2003). On 18 January 2019, nine initial *M. californianus* individuals were hand–collected from the intertidal zone of Horseshoe Cove (38.33325° N, 123.0480571° W) during low tide (–1.13 m) (Fig. 1). Live specimens were collected along a 6 m transect: three specimens from high intertidal position (HIP) 0 m from shore, three specimens from middle intertidal position (MIP) 3 m from shore,
and three specimens from low intertidal position (LIP) 6 m from shore. Specimens were immediately sacrificed by scraping



soft tissue from the shells. Valves were scrubbed with hydrogen peroxide to remove epibionts, rinsed with deionized water, oven–dried at 40°C for 30 minutes, and air–dried overnight. Additional specimen collections took place on 11 July 2019 and 6 June 2020 at BMR. As previously, shells were hand–collected by intertidal position at Horseshoe Cove and from the open–coast BMR site in order to compare shells from a variety of micro–environments: tidal position (HIP, MIP, LIP) and habitat

type (open–coast or protected). Both BMR collection sites represent fully marine rather than estuarine conditions, with average annual salinity at both sites ranging from 33.3 to 33.6 PSU.

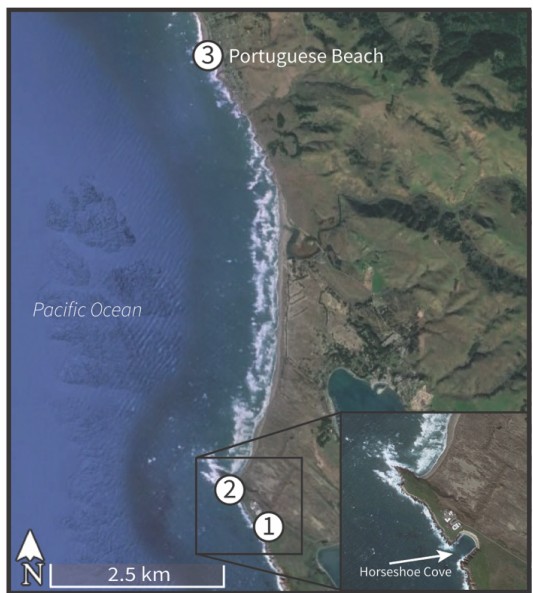

**Figure 1.** Intertidal collection sites at Bodega Bay, California along Sonoma Coast. Sites 1 and 2 are located within Bodega Marine Reserve and Site 3 is located at Portuguese Beach. Modern shells were collected alive in 2019 and 2020 at sites 1 and 2 (Horseshoe Cove

and the open-coast site, respectively). Archival shells were collected alive 7 km north at Portuguese Beach in 2002 and 2003. © Google Earth Pro 2021.

Additionally, 13 archival *M. californianus* shells live–collected at Portuguese Beach on 10 May 2002, 19 July 2002, 1 December 2002, 23 December 2002, and 7 September 2003 were included for temporal comparisons. Portuguese Beach is an open–coast, rocky intertidal site ~ 7 km north of the BMR collection locations. All 13 Portuguese Beach specimens were

collected from the MIP by M.A. Kennedy for dissertation research with the UC Davis Department of Anthropology (Kennedy, 2004).

A thin section was prepared from one valve of each *M. californianus* specimen. Valves were cut along the axis of maximum growth (Fig. 2) using a Buehler Isomet saw with a 0.3 mm diamond wafering blade. Shell cross sections were mounted to an extra–large (50 x 75 mm) glass slide with epoxy and cured at 80°C. Cross sections were polished with a

Buehler Petro–Thin saw and then polished repeatedly using diamond suspension, colloidal alumina suspension, and





microcloth until each shell cross section was polished to a uniform thickness of 300 μm. Shell thin sections were immersed in Mutvei's solution at a temperature of 37°C for 15 minutes under constant stirring (Schöne et al., 2005b). Mutvei's solution stains organic–rich material with blue pigment and exposes mineralized growth increments, revealing the prismatic or tabular crystallographic microstructure of each calcium carbonate layer (Schöne et al., 2005b). After treating samples with
Mutvei's solution, each thin section was rinsed with deionized water and air–dried overnight.

**2.3 Analysis of shell characteristics**

Six shell characteristics were measured in each specimen: shell length (as shown in Fig. 2), maximum valve cross–sectional thickness, the thickness of the innermost calcium carbonate layer, the color of the final growth band, the width of the final growth band, and the standardized gray–value variance as a proxy for growth band contrast. The shell length of each whole
valve was measured parallel to the hinge using digital calipers (0.1 mm accuracy) to estimate the relative ontogenetic age of each individual (i.e., longer shells lived longer), although no time–calibrated estimate of age from shell length exists for northern *M. californianus* individuals. Thin sections were examined both before and after Mutvei's treatment with an Olympus BH–2 light microscope and attached camera with ScopePhoto software. Using Fiji imaging software (formerly ImageJ, available at https://imagej.net/Fiji), each shell's cross–sectional thickness was measured digitally near the umbo at
the region of interest (Fig. 2b). Thin sections were photographed for analysis in Fiji in order to identify mineralogical layers and quantify (count and measure) dark–light growth bands. The thickness of the innermost calcium carbonate layer was also measured digitally at the region of interest near the umbo from oldest to youngest shell material (Fig. 2b). Photomicrographs taken of the region of interest were converted to 8–bit images for each specimen. To supplement visual inspection, gray values were obtained from the 8–bit image through a transect of dark-light banding at the region of interest. To determine the
percent of light bands in each individual specimen, gray values greater than the mean were considered light bands, and gray values less than the mean were considered dark bands. We also calculated and standardized each specimen's gray–value variance as a proxy for band expression; a higher standardized gray–value variance indicated greater contrast between dark–light bands and, while low standardized gray–value variance indicated weak contrast between dark–light bands.





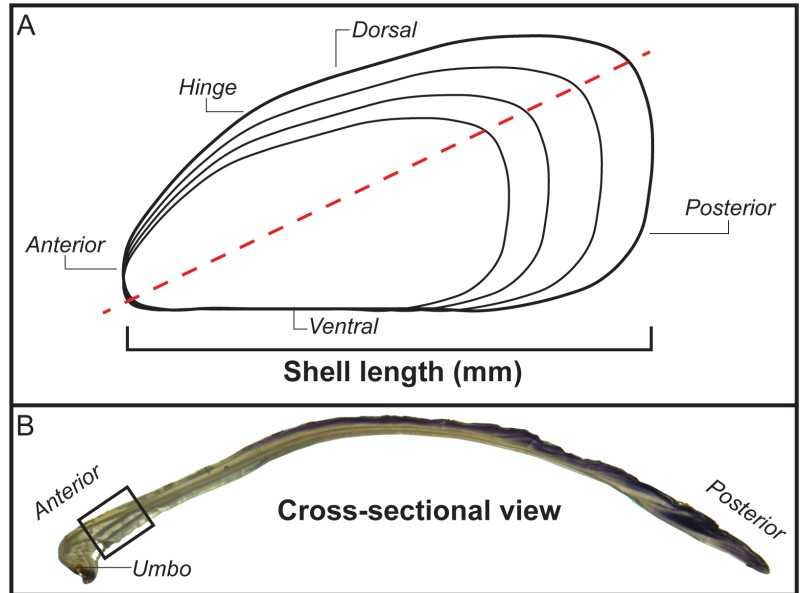

**Figure 2.** Anatomy of *M. californianus* whole valve and cross section. (a) Line drawing of whole valve showing where shell length was measured. Red dashed line denotes maximum growth axis along which the valve was cut to prepare a thin section. (b) Cross-sectional photograph of a shell taken under light microscope. Black box denotes region of interest where the maximum valve cross-sectional thickness, the thickness of the innermost calcium carbonate layer, the color of the final growth band, and the width of the final growth band were measured or noted for all specimens.

Shell growth features were analyzed statistically using the Kruskal–Wallis rank sum test to compare shell thickness to length ratios, Welch *t*–test to compare gray–value variance and percent of light bands across specimens, and Pearson's chi–square test with Yates' continuity correction to assess relationships between terminal band color and collection season. All statistical analyses were performed in R.

**2.4 Analysis of environmental data**

We accessed daily Bodega Bay SST data for the decades 1995–2004 and 2011–2020 provided by the Bodega Ocean Observing Node (BOON) ShoreStation Seawater Observations from University of California, Davis Bodega Marine Laboratory. Since warm (or cool) conditions occur synchronously during the same weeks or months along the Sonoma Coast, BOON data are regionally representative of all three collection sites. SST datasets were averaged to generate monthly, seasonal, and annual mean temperature profiles. Daily SSTs for the 30 days prior to mussel collection dates were plotted for all eight collection dates to examine temperature conditions over the final month of each individual's lifespan.





Upwelling conditions for the same periods were assessed using the Coastal Upwelling Transport Index (CUTI) and Biologically Effective Upwelling Transport Index (BEUTI) for 38° N (Jacox et al., 2018). CUTI represents the rate of

vertical water volume transported per second per meter of coastline at each 1° of latitude along the U.S. West Coast and incorporates impacts of Ekman pumping and cross–shore geostrophic flow (Jacox et al., 2018). CUTI was used as a measure of physical upwelling strength. BEUTI, a measure of vertical nitrate flux (Jacox et al., 2018), was used as an indicator of productivity in the surface waters. BEUTI and CUTI are typically positively correlated, but both indices were used and compared here since physical water transport and nutrient flux can become decoupled in the CCS during alongshore

advection or anomalous oceanographic events (e.g., coastal–trapped wave propagation) (Jacox et al., 2018; Renault et al., 2016). Additionally, both CUTI and BEUTI were considered in case of disproportionate or separate influences of each metric on mussel shell growth.

Both daily data and a 14–day running mean were plotted to characterize environmental conditions for all three datasets (SST, BEUTI, CUTI) for the study periods 1995–2004 and 2011–2020 (Figs. S1, S2 in the Supplement). We

calculated the standard deviation (σ) and plotted y = ± 1σ for SST records and y = ± 2σ for upwelling records to approximate typical ranges of variability for each decade–long study period. We chose to examine 10 year–long windows of time at daily resolution for three reasons: (1) to gauge intra–annual and interannual environmental variability at the study area, (2) to account for the decadal scale variability of PDO, and (3) to examine environmental conditions over the typical lifespan of intertidal *M. californianus*. While the full lifespan of *M. californianus* is unknown, individuals have been known to live up to

11 years (McCoy et al., 2011; Pfister et al., 2011) and even hypothesized to be capable of surviving 50–100 years in undisturbed settings (Suchanek, 1981), although this has not been tested or documented in the literature. We chose ocean records spanning a decade over the years that these shells were collected to provide reasonable environmental context for shell growth patterns for individuals of various and unknown ages.

In addition to examining SST data over daily, monthly, seasonal, and annual scales, we also calculated the

cumulative average SST of each month for all years in each study period (e.g., all January months over 1995–2004) to characterize the annual temperature cycle at Bodega Bay for each decade (Fig. S3). To assess any changes in SST and upwelling between the two decade long study periods, we performed a Two–sample $t$–test to identify any significant differences between means and an $F$–test of equality of variances. All oceanographic data were analyzed and plotted in R.

## 3 Results

### 3.1 Shell characteristics: mineralogical layering

When examined under light microscope, all 40 Bodega Bay *M. californianus* specimens (n = 13 from 2002 and 2003; n = 27 from 2019 and 2020) exhibited three mineralogical layers: outer prismatic calcite, a thin middle layer of nacreous aragonite, and a secondary inner layer of calcite (Fig. 3). The outer prismatic layer made up each shell's exterior, coated by a protective



protein–rich periostracum. The periostracum was partially or mostly worn away from wave exposure in all of our specimens.

The outermost calcite layer grows as ventral margin extension, adding to the shell length by terminal accretion, as evidenced by the direction of the faint, thin growth bands in this layer (Fig. 3c). The outer calcite layer was the only layer to extend consistently throughout the shell from the umbo to the commissure. The aragonite layer appeared to cut through the middle of the cross–section, separating the inner and outer calcite layers (Fig. 3b). The composition of the umbo was primarily aragonite, although the proportion of aragonite to calcite in the umbo varied across specimens.

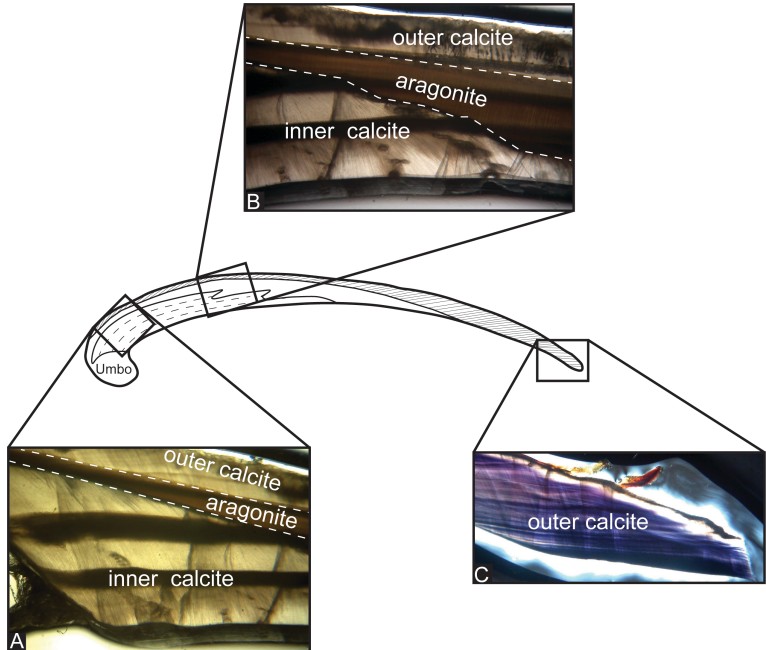

**Figure 3.** Shell structure and mineralogical layering in *M. californianus*. (a) Photo taken under light microscope focused at the region of interest showing the three mineralogical layers. Dashed lines indicate boundaries between inner calcite, aragonite, and outer calcite layers. (b) Photo taken towards the middle of the cross section showing that the inner calcite layer tapers to an end. (c) Photo taken at the posterior margin of the cross section at the commissure, where the outer calcite layer is the only layer present.

In thin section, the aragonite layer was visually distinct from the fan–like prisms characteristic of biogenic calcite. Faint banding did appear in portions of the aragonite layer; in some specimens, the banding appeared continuous with the dark–light banding in the inner calcite layer (Fig. 3b). The nacreous aragonite layer began at or near the umbo toward the outer margin of the shell but did not extend all the way to the commissure in any specimen. Under plane polarized light, the aragonite layer appeared brown in color and contained tabular crystals oriented parallel to the shell's surface. The innermost

layer of prismatic calcite near the anterior margin was microstructurally similar to the outer calcite layer, although the inward growth direction of the inner calcite layer adds to the shell thickness rather than shell length. The inner calcite layer





was also the only layer to contain thick, strongly expressed dark–light band pairs (Fig. 3a). The most recently formed growth band, or terminal band, is the innermost band distal to the outer calcite layer.

Shell morphology is controlled in part by the growth rates of each mineralogical layer, since the inner calcite layer

contributes mainly to valve thickness and the outer calcite layer contributes mainly to shell length. There was a statistically significant positive relationship between the inner calcite layer and shell length (Linear regression, $R^2 = 0.36$, $p < 0.001$, $F_{1,38}$ = 21.49, Inner calcite thickness (mm) = 0.03 * Shell length (mm) – 0.06)).

Thick dark–light growth band pairs were present in the inner calcite layer (Fig. 3a) and faint, indistinguishable bands appeared in the outer calcite layer (Fig. 3c). The inner calcite layer of all specimens contained an average of three

growth band pairs, ranging from 0 to 10 pairs per specimen, while the bands in the outer calcite layer were unquantifiable due to their faint and inconsistent expression. Growth band contrast (visual distinction between dark and light banding) and pattern (number of band pairs, color of terminal band, and band thickness) varied widely across specimens (Table S1). Standardized gray–value variance was used as a proxy for growth band contrast. Mean standardized gray–value variance of all specimens (n = 40) was 0.0 with a standard deviation of 1.0. High standardized gray–value variance is a quantitative

indicator of a high contrast between dark and light bands, interpreted as strongly expressed banding. Conversely, low standardized gray–value variance is an indicator of low dark–light band contrast, interpreted as weakly expressed banding (see Fig. S4 for examples of high and low contrast banding).

To determine whether inner calcite dark–light banding continues to form throughout ontogeny, we used shell length as an indicator of relative ontogenetic age. We applied reduced major axis (RMA) regression to assess the relationship

between dark–light band pairs and shell length (Fig. 4). We found weakly positive and statistically significant correlation between shell length and dark–light band pairs with greater variance among larger (older) individuals (RMA regression, $R^2$ = 0.39; $p < 0.001$, Number of dark–light pairs = 0.14 * shell length (mm) – 4.9).


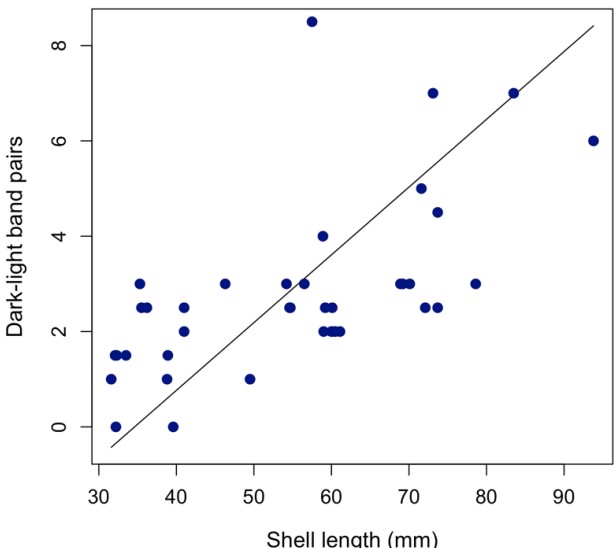

**Figure 4.** Relationship between shell length and number of growth band pairs for all 40 shells with an RMA regression line plotted;
Number of dark-light pairs = 0.14 * shell length (mm) – 4.9.


Across all specimens, there was no statistically significant relationship between standardized gray–value variance
(an indicator of growth band contrast) and shell length (Pearson's correlation, p = 0.15). Shell characteristics for all
specimens are provided in Table S1.

**3.2 Micro–environment and growth band contrast**

Standardized gray–value variance varied depending on the micro–environment (Fig. 5). Growth band contrast was first
compared among specimens collected from a protected cove environment (Horseshoe Cove) and the open–coast sites (BMR
and Portuguese Beach) (Fig. 5a). Specimens from open–coast habitats had a lower and broader range of gray–value variance
(mean ± σ = 0.37 ± 1.26, n = 17) than cove specimens (mean ± σ = – 0.27 ± 0.66, n = 23). Nearly all specimens with high
growth band contrast (standardized gray–value variance > 1) were collected from an open–coast habitat, although this
relationship was not statistically significant (Welch two–sample *t*–test, t = – 1.91, p = 0.07).

Specimens were also categorized by intertidal position (LIP, MIP, HIP) (Fig. 5b). All specimens collected from LIP
and HIP had low gray–value variances (ranging from –1.1 to 0.59, n = 13). Specimens with the highest contrast in gray
values were collected from the MIP (ranging from –0.97 to 2.88, n = 27). Even when archival shells were excluded (all
archival shells were collected from MIP), the modern shells displayed the same patterns, with the greatest range and highest
standardized gray–value variance still found in MIP specimens only (Fig. S5).

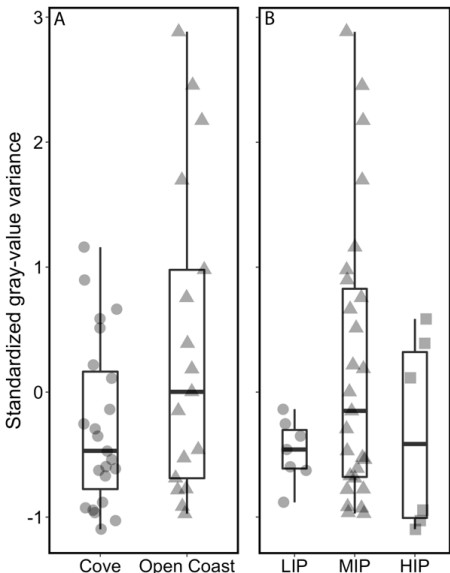

**Figure 5.** Relationships between micro-habitat and standardized gray-value variance as a proxy for growth band contrast. High standardized gray-value variance is indicative of high contrast between dark and light bands. (a) Cove specimens are from Horseshoe Cove at BMR and open-coast specimens are from an open-coast site within BMR and Portuguese Beach. (b) Relationships between intertidal position (LIP = low intertidal position, MIP = middle intertidal position, HIP = high intertidal position) and standardized gray-value variance for all 40 specimens.

### 3.3 Oceanographic conditions and growth band pattern

Thirteen specimens precipitated a light terminal band. Out of these 13 specimens, 10 were collected during months with average monthly SST between 12.75 and 13.5°C (Fig. 6a) and mean aggregated seasonal SST of 12°C (Fig. 6b). Out of all 40 shells, 27 shells had a dark terminal band, and 24 of these were collected during months with either monthly SST lower than 12.75°C or higher than 13.5°C. All six specimens collected during a month with monthly average SST ≤ 12°C had a dark terminal band, and 22 out of 24 specimens collected during months cooler than 12.75°C had a dark terminal band (Fig. 6a).


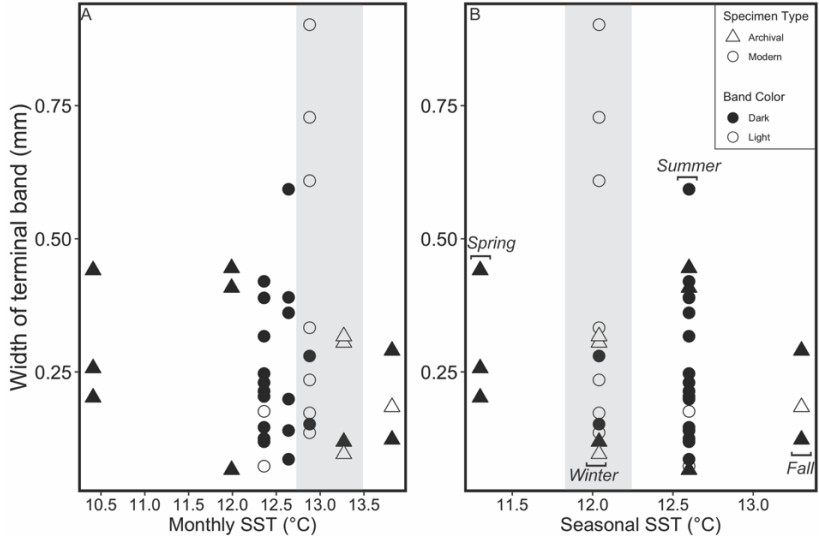

**Figure 6.** Relationships between SST, width of terminal band, color of terminal band, and specimen type for all 40 specimens. Legend applies to both plots. Band color refers to the color (dark or light) of the terminal band. (a) Relationships between mean monthly SST during month of collection, terminal band color, and terminal band width. Shaded bar represents approximate monthly SST range over which most specimens are associated with light terminal bands (12.75-13.5°C). (b) Relationships between mean cumulative seasonal SST during season of collection, terminal band color, and terminal band width. Shaded bar represents seasonally averaged SST (and therefore season) with which most light terminal bands are associated. Seasons are labeled on plot.

In addition to monthly and seasonal SST patterns, daily SST for the final 30 days of each individual's life were plotted (Figure 7). Collection dates in December 2002 were preceded by extremely steady daily temperatures between 12.5 and 13°C (n = 4, and three of these specimens had a light terminal band) (Figs. 7c, d). Oscillating warm-cool daily temperatures occurred in July 2002, September 2003, and June 2020 and were closely associated with dark terminal bands (Figs. 7 b, e, h). An extreme warm spike to 20°C occurred over the course of three days in July 2019 (Fig. 7g) and all six specimens collected following this event had a dark terminal band. Instrumental error occurred in May 2002, so it was not possible to connect the three specimens with dark terminal bands collected during that month with daily temperature trends (Fig. 7a). Seven out of nine specimens collected in January 2019 had a light terminal band despite a three day temperature spike from 12.5°C up to ~ 16.25°C (Fig. 7f).

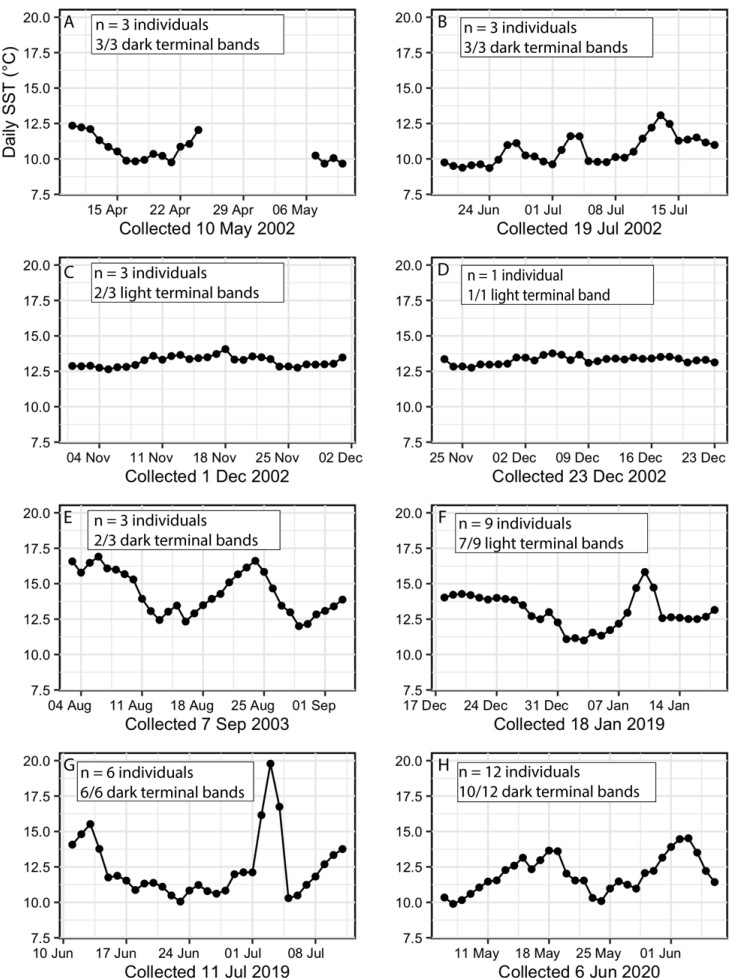

**Figure 7.** Daily temperatures for the 30 days prior to the collection date for all 40 specimens. Each plot features the number of specimens collected on that date and the majority terminal band color. (a) Instrumental error occurred in May 2002. (b) Daily temperatures oscillating between cooler (~ 9°C) and warmer (~ 13°C) in July 2002. (c, d) Consistent daily temperatures between 12.25°C and 12.5° in December 2002. (e) Sinusoidal daily temperatures oscillating between ~12° and 17.5°C in September 2003. (f) Daily temperatures prior to 18 January 2019 collection date were consistent for two weeks before slight cooling and three days of warming. (g) Highly variable daily temperatures and a three day extreme warm spike in July 2019. (h) Daily temperatures oscillating between cooler (~ 10 °C) and warmer (~ 15 °C) in June 2020.

Terminal band color and width were also assessed in relation to upwelling conditions expressed as BEUTI and CUTI. Ten out of 13 shells with light terminal bands appeared in shells collected during months with negative BEUTI and CUTI values. Out of the 27 shells with a dark terminal band, 24 were collected during months where CUTI values were 0.5





or greater and BEUTI values were greater than 5.0. Patterns were similar for both BEUTI and CUTI (Fig. S6.) and strongly covaried with monthly SST (i.e., intense upwelling produces cool conditions while relaxed upwelling produces warm

conditions).

**3.4 Temporal trends of shell growth features**

On a sub–annual scale, dark–light band pairs in the inner calcite layer displayed a strong relationship with season of collection in both modern shells (n = 27, collected in 2019 and 2020) and archival shells (n = 13, collected in 2002 and 2003). Of the 21 mussels collected in summer months with increasing temperatures, 19 had a dark band that precipitated as

the terminal band (90.47 percent). Of the 13 mussels collected in winter months with stable temperatures, 10 had a light band that precipitated last (76.9 percent). Only six mussels were collected during spring and fall (n = 3 each), but all of the shells collected in the spring have a dark band that precipitated as the terminal band and two out of three of the shells collected in fall have a dark band that precipitated as the terminal band (Table S1). We identified a statistically significant relationship between season of collection and terminal band color (Chi–square test, $\chi^2$ = 18.193; df = 3, p = 0.0004).

Decade–specific growth trends were assessed by comparing shell characteristics between archival and modern shells. We compared gray–value variance measurements, inner calcite layer thickness to valve length ratios, and percent of light bands per shell (Fig. 8). The standardized gray–value variance was significantly lower in modern shells than in the archival shells (Welch two–sample $t$–test, t = 2.27; df = 14.68; p = 0.039; Fig. 8a). The standardized ratios of the inner calcite thickness relative to the shell length were significantly lower in modern shells than they were in shells collected in

2002 and 2003 (Kruskal–Wallis rank sum test, p < 0.05) (Fig. 8b).

There was no statistical difference between the percent of light bands in archival versus modern shells (Welch two–sample $t$–test, p = 0.5) (Fig. S7). The percent of light bands ranged from 0.15 to 0.61 in all specimens, with a mean of 0.43 and a median of 0.46.





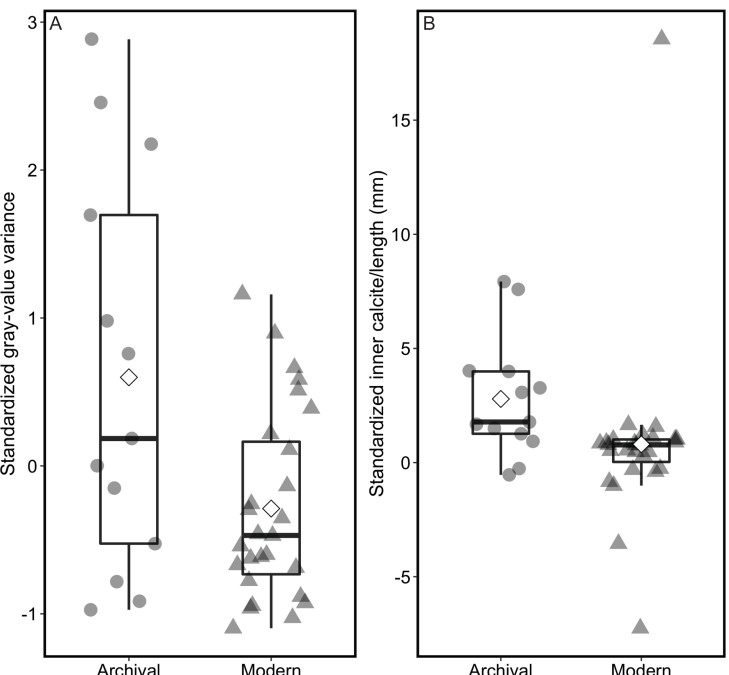

**Figure 8.** Box plot showing the range of standardized ratio of inner calcite thickness to shell length in archival and modern specimens. White diamond denotes mean.

### 3.5 Evaluation of oceanographic data

Using BOON SST data for Bodega Bay and monthly upwelling indices at 38° N from BEUTI and CUTI, we assessed daily, monthly, seasonal, and annual conditions for the study periods 1995–2004 and 2011–2020 to provide environmental context for the archival and modern shells, respectively. Over both study periods, the lowest SSTs occurred in April through June and the warmest months of the year are August through November (Fig. S3). Temperatures change drastically from April through July, with SST shifting from one annual extreme (< 10° C) to another (~ 16° C) within a three month period (Fig. S1). The coolest season of both study periods was spring of 2002 while the warmest season was winter of 2002 (Table 2). Summers recorded higher daily temperature variability ($\sigma > 1.5$) than winter and spring months ($\sigma < 1$).




**Table 2**. Seasonally averaged SST (°C) for the specific collection season of each mussel (n = 40 over eight collection dates across seven different seasons) and standard deviation for each calendar season of the study period. Spring = March through May; summer = June through August; winter = November through January, fall = September through November.

| Season | Mean seasonal SST (°C) | σ |
|---|---|---|
| Spring 2002 | 10.9 | 0.88 |
| Summer 2002 | 11.8 | 1.72 |
| Winter 2002 | 13.2 | 0.38 |
| Fall 2003 | 12.6 | 1.09 |
| Winter 2019 | 13.1 | 0.93 |
| Summer 2019 | 13.1 | 1.83 |
| Summer 2020 | 13.1 | 1.73 |


Comparing the two study periods revealed no significant difference in mean monthly or annual SST values between the 1995–2004 period and 2011–2020 period (Two–sample $t$–test, p = 0.12 and 0.58, respectively). However, greater variance in SST occurred in the more recent study period (F–test, F = 0.6499; $df_{num}$ = 3140, $df_{denom}$ = 3630; p < 0.001). Mean annual SST was at or below 12.5°C from 1999 to 2004, and near or above 12.5°C from 2014 to 2020 (Fig. 9).

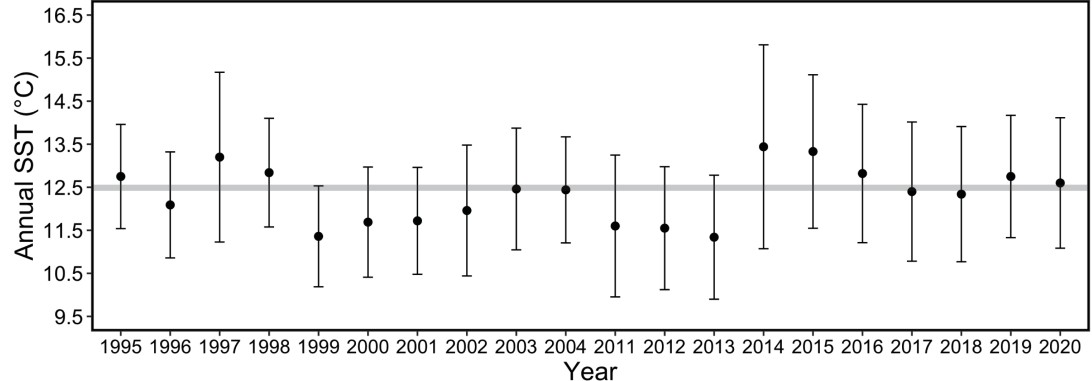


**Figure 9.** Mean annual SST for each year of the study period. Error bars represent standard deviation of daily temperatures over the course of each year. Shaded bar plotted at 12.5°C for ease of visual comparison across annual means from 1995 to 2004 and 2011 to 2020.

Both the archival and modern upwelling indices record weak upwelling during winter with low productivity (Fig. S8). Spring and early summer (March through June) is characterized by strong upwelling and high productivity, reflected by

high BEUTI and CUTI values. Comparisons between upwelling indices revealed a significant difference in mean monthly CUTI values (Two–sample $t$–test, t = –3.3339; df = 7669; p = 0.0008) and variance values (F–test, F = 1.1503; $df_{num}$ = 4017, $df_{denom}$ = 3652; p < 0.001) with higher averages and greater variance occurring in the more recent study period (Fig. S2). The





same shift was present in the BEUTI record for both average values (Two–sample $t$–test, t = –5.5299; df = 7669; p < 0.001)
and variance (F–test, F = 0.902; $df_{num}$ = 4017, $df_{denom}$ = 3652; p = 0.001), with significantly higher means and greater

variance of BEUTI values in 2011–2020 than in 1995–2004.

## 4 Discussion

### 4.1 Interpreting mineralogical layering and growth bands

Visual inspection under microscope showed three distinguishable mineralogical layers in Bodega Bay specimens (Fig. 3),
regardless of collection location or shell length. The mineralogical layering of *M. californianus* was first qualitatively
described from visual inspection six decades ago (Dodd, 1964) and had not been re–examined in the literature prior to this
study, leading to inconsistencies in the *M. californianus* literature. While all *Mytilus* congeners precipitate both calcite and
aragonite in two distinct layers in their shell (Taylor et al., 1969), *M. californianus* is the only *Mytilus* species known to
precipitate a secondary layer of calcite. A few previous *M. californianus* studies (e.g., McCoy et al. 2011, 2018; Pfister et al.
2011, 2016) noted the presence of a secondary inner layer of calcite, as initially described by Dodd (1964) and corroborated
by the specimens analyzed here, but *M. californianus* shell mineralogy is often described or assumed to be bi–layered, with
an outer calcite layer and an inner aragonite layer only.

        The dark–light bands within the inner calcite layer of *M. californianus* have a different appearance from growth
lines in many other well–studied bivalve species. Growth bands in *M. californianus* are thick bands alternating between dark
and light increments, rather than thin lines that demarcate periods of accretionary growth. For example, in *Arctica islandica*
and *Saxidomus gigantea*, thin growth lines result from growth cessation and reliably represent growth shutdown during
winter, while light increments represent shell growth over the rest of the year (Schöne et al., 2005a; Hallmann et al., 2009;
Burchell et al., 2013). Growth lines are understood to represent points at which calcification ceased and subsequently
resumed, but formation of dark–light banding is more complex. The dark–light bands in *M. californianus* are more
comparable to those found in *Crassostrea virginica*, which are described as alternating dark and light increments visible in
cross–section (Kirby et al., 1998; Andrus and Crowe, 2000; Surge et al., 2001; Zimmt et al., 2019; Table 1). Dark–light
bands have been suggested to represent alternating periods of shell deposition during aerobic and anaerobic respiration, with
light bands forming during aerobiosis and dark bands forming during anaerobiosis (Lutz and Rhoads, 1977; McCoy et al.,
2011). Under optimal conditions, such as during immersion or moderate to warm temperatures, *M. californianus* gapes and
respires aerobically, and during sub–optimal conditions, such as during aerial exposure and/or extreme temperatures, *M.*
*californianus* closes its valves and respires anaerobically (Bayne et al., 1976; Connor and Gracey, 2011; Connor et al.,
2016). During anaerobiosis, glucose and aspartate ferment, resulting in the production of alanine and succinate, which is then
converted to propionate if anaerobic conditions persist for several days (Connor and Gracey, 2011). Succinate is an acidic
end–product of anaerobiosis, which may contribute to the production of an organic–rich dark band in the shell's inner calcite
layer (McCoy et al., 2011). While it has been suggested that *M. californianus* partially dissolves its own shell during





anaerobiosis in order to neutralize the acidic end–products of its own metabolism (McCoy et al., 2011), the mechanism that produces dark bands in *M. californianus* has not been identified; dark bands could be dissolution bands, organic–rich bands, or simply the visual expression of slow calcite biomineralization during sub–optimal growing conditions.

It also remains unclear if one pair of dark–light bands reliably represents one year in Bodega Bay specimens, as has been documented in populations of *M. californianus* from Tatoosh Island and Seattle, Washington (McCoy et al. 2011, 2018;
Pfister et al. 2011, 2016). If dark bands in the inner calcite layer were to form annually in response to temperature cycles, then individuals of the same size class (and therefore age cohort) should have the same number of dark–light band pairs. Shell growth rate monitoring of *M. californianus* individuals from Bodega Bay and other northern California coast sites has shown that individual shells < 80 mm long can grow between 0 and 1 mm per month (Smith et al., 2009). A young individual with an initial size of 10 mm growing 1 mm per month would grow ~ 30 mm in 30 months, or 2.5 years. This
individual would be 40 mm long with 2.5 dark–light pairs after 2.5 years. While multiple specimens analyzed here did follow these estimates (Fig. 4), many individuals contain far fewer dark–light band pairs than their shell length would indicate, so it would not be possible to visually cross–date, as can be done in other bivalve species (e.g., *Panoptea abrupta*) (Black et al., 2008).

The statistically significant positive correlation between shell length (a relative proxy for ontogenetic age) and
dark–light band pairs within the inner calcite layer (RMA regression, $R^2 = 0.39$, $p < 0.001$; Fig. 4) suggests that *M. californianus* shells continue to form dark–light bands in the inner calcite layer as the shell grows posteriorly (i.e., the inner calcite layer thickens as the outer calcite layer lengthens throughout ontogeny). However, constraining calcification rates for *M. californianus* from northern California is complicated by slow shell growth rates (Smith et al. 2009) and the lack of a reliable ontogenetic age estimate based on shell length. Previous shell growth studies have focused on the fast–growing
southern California *M. californianus* populations (Blanchette et al., 2007; Smith et al., 2009; Ford et al., 2010; Connor and Robles, 2015). An early study conducted eight decades ago used 1000 mussels growing in La Jolla, California to estimate that an average *M. californianus* shell is ~ 80 mm long after one year (Coe and Fox, 1944). However, the latitudinal gradient and local oceanographic differences are strong indicators that the shell length–based age estimate for southern California mussels is not applicable to northern California mussels, where seawater is significantly cooler and upwelling is more
intense. In addition to geographic and oceanographic differences, environmental changes that have occurred in the ~ 80 years since the study necessitate an updated and site–specific estimate of *M. californianus* shell growth rate. Extremely low shell ventral margin extension rates have been reported for *M. californianus* at Bodega Bay (~ 0 mm per month) (Smith et al., 2009) and the coast of Washington (~ 1 mm per month) (Paine, 1976). We found further evidence for markedly lower growth rates for northern *M. californianus* shells as a result of our collection methodology (Text S1). The disparity between
shell growth rates for northern and southern California mussel populations also influences the age at sexual maturity (Suchanek, 1981). Individuals from southern California reach sexual maturity at ~ 15–25 mm long or approximately four months after settlement (Coe and Fox, 1944; Jones and Richman, 1995), but the shell length and age at sexual maturity are unknown and certainly different for slow–growing northern California mussel shells.



### 4.2 Relationships between the environment and growth band patterns

We observed relationships between micro–environment and growth band contrast, with open-coast and MIP specimens containing more strongly expressed dark–light bands than specimens collected from cove or extreme (LIP or HIP) tidal environments (Fig. 5). High standardized gray–value variance was again found only in MIP specimens even when all archival shells were excluded from analysis (since all archival shells are from the MIP at Portuguese Beach; Fig. S5). The similarity in gray–value variance patterns across all sets of MIP specimens indicates that growth band contrast is more

strongly controlled by subtle differences in micro-environment (e.g., aerial immersion time) than by the alongshore coastal oceanographic gradient of the Sonoma Coast. Such differences emphasize the importance of small–scale, within–site variation of calcification patterns for *M. californianus*, as previously presented by Thakar et al. (2017) for regions like Sonoma Coast with locally uniform oceanographic patterns. However, in regions with high alongshore variability or locally asynchronous warm/cool periods, immediate differences in local oceanography would also play an important role in

influencing calcification patterns. For example, there are significant differences in *M. californianus* shell growth rates just north and south of Point Conception in southern California, where SSTs and wave exposure vary strongly despite geographic proximity (Blanchette et al., 2007).

In addition to micro–environmental variation, we found relationships between broader oceanographic conditions and shell growth features. Shells collected during months with strong upwelling and high productivity were more likely to

have a dark band that precipitated most recently (Fig. S6), indicating that strong upwelling and resultant high food availability does not cause faster shell growth in *M. californianus* from northern California. Low temperatures induced by upwelling may outweigh the effects of high food availability on calcification rate. Temperature – rather than upwelling and food availability – has been previously identified as the primary factor of shell growth rate for *M. californianus* populations in southern California in field studies (Phillips, 2005; Blanchette et al., 2007). We contribute an additional line of evidence

for SST as the strongest control over growth rate in both archival and modern northern *M. californianus* shells and we suggest that SST stability (or variability) has a stronger influence on growth band coloration and growth rate than absolute temperature (Fig. 6). While previous studies have suggested that food availability is not a strong driver of calcification for *M. californianus* because of the lack of relationship between chlorophyll–*a* and shell growth (Phillips, 2005; Blanchette et al., 2007), it is also possible that upwelling–associated low pH can expose mussels to OA conditions in their habitats (Feely et

al., 2008; Rose et al., 2020). Periods with heavy upwelling in the CCS could hinder calcification or even alter the proportion of shell calcite to aragonite in response to changes in the calcium carbonate saturation state (Bullard et al., 2021). While the heavy upwelling in the CCS offers a natural laboratory for examining low pH conditions, it may become increasingly difficult to disentangle the impacts of anthropogenic OA and upwelling–induced low pH on *M. californianus*, given that wind–driven upwelling has increased along the coast of California in recent decades due to intensifying onshore–offshore

atmospheric pressure gradients associated with rising temperatures (García–Reyes and Largier, 2010). Evidence of decadal scale intensification of upwelling appeared here in the CUTI and BEUTI records.





### 4.3 Interpreting temporal trends of growth features

Using the season of collection for each shell, we identified a significant relationship between season and band color, and
specifically between winter and light bands (Fig. 6b). While winter is typically the time of year that bivalves in the northern
hemisphere experience growth slowdown and produce a dark line (Killam and Clapham, 2018, and references therein), our
findings from *M. californianus* support the hypothesis that mussels grow their shell optimally during warmer periods (up to a
point) given that December at Bodega Bay has higher mean monthly SST and more stable daily SSTs than March through
June, making winter months generally warmer than spring months and much less variable during summer months (Table 2).
We interpret the light band found in the majority of winter–collected shells as an indicator of fast (calcium carbonate–rich)
growth occurring during optimal growth conditions for northern California mussel populations when SST is moderate and
stable. Dark (slow–growth) bands were closely associated with spring–early summer, or during cooler or highly variable
temperatures (Fig. 7). Dark bands were also found in all specimens that experienced an extreme three day long heat wave
(20°C) that occurred a week prior to their collection (Fig. 7g), indicating that *M. californianus* growth rate slows during
variable, cold, or extremely high SSTs. Despite the strong correlation between season and band color, dark–light bands
cannot necessarily be used as an indicator of lifespan because the dynamic oceanographic regime at Bodega Bay could result
in multiple growth slowdowns within the span of one annual cycle. However, dark bands could potentially be used to
reconstruct extreme or variable conditions while light bands could serve as indicators of stable or moderate periods.

In addition to seasonal variability at Bodega Bay, the intertidal zone experiences extreme environmental variation
(temperature, submergence time) on daily and bi–weekly scales. While tidal cycling does not contribute to the number of
dark–light bands in the inner calcite layer (i.e., mussels experience hundreds to thousands of high–low tidal cycles but
contain only three dark–light band pairs on average), tidal variability could play a role in growth band contrast since
intertidal position (LIP, MIP, HIP) and standardized gray–value variance were related (Fig. 5). LIP and HIP specimens
experience tidal extremes for the longest periods of time (immersion and exposure, respectively) and have low contrast
growth bands. MIP specimens have a wide range of standardized gray–value variance, but all specimens with strongly
expressed bands were collected from the MIP only, perhaps because they experienced tidal extremes for shorter periods of
time than LIP and HIP specimens.

We calculated the percentage of light bands across all specimens to compare the time spent growing normally (light
bands) versus abnormally (dark bands). The average percent of light bands across all specimens was 43 percent and no
specimen had a light band percentage greater than 61 percent (Fig. S7). If dark bands precipitate more slowly than light
bands yet represent half or more than half of the inner calcite layer of all specimens, then Bodega Bay specimens spend more
of their lives experiencing hindered growth rather than normal growth. Interestingly, northern California mussel populations
seem to grow their shell during faster "growing windows" when conditions are moderate to warm and stable, but they calcify
slowly for a longer period of the year and spend most of their lives experiencing slow–growth conditions. While most light



bands are associated with winter (December and January), it is possible that the warmest time of year at Bodega Bay (~ August through October) could also be part of the "growing window" for northern California populations of *M. californianus*, but we did not have enough shells collected in fall months available (n = 3) to confirm this. Out of the three fall–collected shells, one specimen did have a light band that precipitated most recently, which does suggest that fast growth and light band precipitation can occur during the warm fall months, or during any period of the year with sustained

conditions that match the optimal range for shell growth. At Bodega Bay, sustained optimal growth conditions are less likely to occur during spring and early summer because of cold and highly variable SST conditions controlled by the seasonal upwelling regime.

        We also observed a statistically significant decline in inner calcite thickness–to–shell length ratios from the archival to the modern specimens, indicating that mussel shells growing in the same location nearly two decades apart are thinner

relative to their length (Fig. 8b). Both the cross–sectional valve thickness and the thickness of the inner calcite layer have declined overall relative to shell length. The valve thinning could indicate a slowed rate of calcification, or that the inner calcite layer now grows for a shorter period of time in the life of the animal, or even that the length of life is declining in modern specimens. Evidence for rapid and recent shell thinning has also been found in Washington *M. californianus* populations, where cross–sectional shell thickness in modern mussels is significantly thinner than both archival (collected in

the 1960s–1970s) and archaeological mussel shells (~ 2420–1000 cal BP) (Pfister et al., 2016). In addition to shell thinning, we found that growth bands in modern shells had significantly lower dark–light band contrast than the archival shells (Fig. 8a). This provides a new line of evidence for recent changes in shell microstructure occurring in the past 15 years, in addition to previous evidence of increased crystallographic disorder and increased amorphous calcium carbonate deposition in modern *M. californianus* shells relative to archival (1960s–1970s) and archaeological (~ 2420–1000 cal BP) shells from

Washington (McCoy et al., 2018). Due to limitations in the length and availability of oceanographic data sets, we did not aim to link changes in shell growth and the dark–light band pattern to anthropogenic OA or lower pH in response to increased CCS upwelling, although SSTs were higher and more variable and upwelling was significantly stronger in 2011–2020 than in 1995–2004. We suggest that the weakened growth band expression and decline in inner calcite thickness ratios in the modern shells could be responses to warmer than average conditions and/or low pH conditions associated with stronger

upwelling, which can be further explored by applying well–developed geochemical proxies to reconstruct conditions recorded by mussel shells (e.g., $\delta^{18}O_{[shell]}$ – SST). However, the primary challenge with sampling the inner calcite layer for stable isotope analysis is the fine–resolution sampling required; the inner calcite layer is ~ 2 mm thick (mean of n = 40) and individual bands can be extremely thin (on the order of μm).

### 4.4 Potential factors contributing to variability of shell growth features

Given that temperature and upwelling conditions are nested within temporal trends (interannual variability and periodic oceanographic phases), we expected and found significant differences in shell growth patterns between the archival and the modern shells. We observed a high degree of variability in growth band pattern and inner calcite thickness to shell length



ratios in both modern and archival specimen categories (Fig. 8). Even when standardized, the variance of inner calcite thickness to shell length ratio across all specimens was high ($\sigma^2 = 7.77$), indicating that there is a range of growth rates and

dark–light band formation rate even among specimens experiencing the same environmental conditions. Some variance is likely a product of uneven sampling distributions due to the available archival specimens and restrictions on modern sample collection during the COVID–19 pandemic. Regardless of sample size, a certain degree of growth pattern variability is expected depending on the plasticity among individual organisms and on the micro–environment within and across populations. If dark–light banding is mediated by a physiological response to environmental conditions (e.g., a metabolism

that alternates between anaerobic and aerobic respiration depending on SST), there can be varying physiological responses among individuals due to micro–environmental gradients in SST or immersion time in the highly variable intertidal zone (Connor and Robles, 2015; Thakar et al., 2017). For food availability, tidal position is a minor factor since functional–submergence time (the time required for an individual to gape its valves and effectively filter–feed) is uniform across intertidal positions (LIP, MIP, HIP) for *M. californianus* (Connor et al., 2016). While low tide and low temperatures have

been suggested as conditions that trigger a switch to anaerobic respiration in *M. californianus* (Connor and Gracey, 2011; McCoy et al., 2011), the temperature threshold for anaerobiosis is unknown and is likely to differ depending on the population's latitude and local oceanographic parameters. For example, northern California mussel populations experience cooler conditions and a narrower range of temperatures (~ 10–13.5°C mean monthly SST) than southern California mussels, which have been observed to grow most rapidly between 15 and 19°C (Smith et al., 2009). Temperatures approaching 19°C

could be outside the range of tolerance for populations from northern California given that SSTs rarely exceed 19°C at Bodega Bay; the BOON SST record documents only three days warmer than 19°C over both decade–long study periods. The north–south SST gradient in California may result in slow growth and more frequent dark band formation for northern mussel populations, and optimal growth and light band formation — and perhaps a greater overall percentage of light bands — for warm–water acclimated southern California mussel populations. A latitudinal gradient in shell growth rate, and

therefore growth band pattern, controlled by the CCS' variable upwelling regime is explained by the high plasticity in physiological responses to oceanographic conditions in *M. californianus* (Dahlhoff and Menge, 1996).

**5 Conclusions**

We identified three mineralogical layers in *M. californianus*: an outer calcite layer with faint indistinguishable banding, a thin nacreous aragonite middle layer, and an inner calcite layer that grows inward, precipitating dark–light band pairs. The

improved understanding of shell layering and growth directionality has important implications for paleoceanography and archaeology, which require geochemical subsampling approaches and proxy equations tailored to growth direction and the specific calcium carbonate polymorph. Within the genus *Mytilus*, the inner calcite layer is unique to *M. californianus* and may be a useful layer for the reconstruction of extreme conditions and determination of season of collection due to strongly





expressed growth banding, although the contrast between dark–light bands is variable and dependent on tidal position and

habitat type.

We documented strongly positive and statistically significant correlation among light bands and winter collection months, moderate SST (average monthly SST between 12.75 and 13.5°C and average seasonal temperature of ~ 12°C), and weak upwelling (CUTI and BEUTI < 0) at Bodega Bay, indicating that light bands are more likely to precipitate during "growing windows" with relatively constant SSTs. Slower or halted growth, recorded as dark bands, is more likely to occur

during spring through early summer, or when conditions are highly variable or locally extreme, although it is uncertain which conditions are considered "extreme" for intertidal *M. californianus* populations in northern California. We also found that low temperatures may result in slow shell growth and dark band formation even during periods of upwelling–induced productivity. Interestingly, most specimens analyzed here contained a greater percentage of dark bands than light bands, suggesting that the growth slowdown period is longer than the "growing season" for Bodega Bay mussels and that mussels

spend more of the year – and more of their lives – experiencing hindered growth rather than normal growth.

A shift in calcification patterns from archival (2002–2003) to modern (2019–2020) mussels is also documented here. The statistically significant decline in growth band contrast and inner calcite thickness to shell length ratios indicates that *M. californianus* is growing more slowly or calcifying less in 2019–2020 than this species was less than two decades ago at the same location. The spatial and temporal variability of *M. californianus* shell growth from Bodega Bay highlights

the need for future site–specific calibration of growth band patterns and comparisons through time. Given that *M. californianus* is an ecologically important foundation species and its shell appears to respond to and sensitively record environmental changes, analysis of the relationships among shell growth features, environmental conditions (SST, pH, and upwelling), and community ecology should be investigated to assess whether these shifts in calcification patterns will have negative impacts for *M. californianus* and the biologically rich intertidal community that this species supports.




**Data availability.** All shell data collected for this paper are available in the Supplementary Information. BOON data are available online at https://boon.ucdavis.edu/ and all upwelling (CUTI and BEUTI) data are available online at https://oceanview.pfeg.noaa.gov/products/upwelling/cutibeuti and https://mjacox.com/upwelling-indices/.

**Supplement.**

**Author contributions.** VPV, SJC, and TMH conceptualized and designed the project. VPV completed data collection and
wrote the manuscript. All authors contributed to data analysis and editing of the manuscript.

**Competing interests.** The authors declare that they have no conflict of interest.

**Acknowledgements.** This material is based upon work supported by the National Science Graduate Research Fellowship under Grant No. 2036201. We thank Jackie Sones for access and assistance at the Bodega Marine Reserve and Zachary Oretsky for assisting with mussel collection. We thank Ann Russell for providing access to archival mussel shells. All thin
sections were prepared by Greg Baxter. Leslie Garcia helped to calibrate scales for thin section images. John Largier provided helpful input about Sonoma Coast oceanography.

**Financial support.** This research was supported by the Geological Society of America (to VPV), the Cordell Durrell Fund (to VPV), the Mildred E. Mathias Grant (to VPV), and the National Science Foundation (OCE 1832812 to TMH and 2036201 to VPV).

**Review statement.**

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
