# Peer review of "Investigating controls of shell growth features in a foundation bivalve species: seasonal trends and decadal changes in the California mussel"

_Biogeosciences, 2021_

## Author Response (AR1)

Author's Response for:

**Investigating controls of shell growth features in a foundation bivalve species: seasonal trends and decadal shifts in the California mussel**

Veronica Padilla Vriesman[1], Sandra J. Carlson[1], Tessa M. Hill[1]

[1]Department of Earth and Planetary Sciences, University of California, Davis, One Shields Avenue, Davis, CA, 95616, USA

At this stage, we made changes suggested by Referee #1 and Referee #2. Included is our Response to Referee Comment 1 (with point-by-point comments), our Response to Referee Comment 2 (with point-by-point comments), and a marked-up manuscript with Track Changes included. We have also submitted a clean version of the revised manuscript.

We thank the two referees and the Associate Editor for their input.

Here is a list of relevant changes to the manuscript, as addressed in the Responses to Referee Comments. The line numbers correlate to the line numbers in the newest, clean version of the manuscript.

- We moved the background information about dark-light bands in bivalve shells from the last sentence of the Introduction to lines 35-39 in the first paragraph of the Introduction.
- We added the Butler et al. (2013) citation (line 46)
- We adjusted line 47 to say "Schöne et al. (2005a) and Schöne (2013) concluded that *A. islandica* produces annual and daily growth lines that form continually throughout the year."
- We added the Wanamaker et al. (2008) citation to the Mg/Ca row of Table 1
- We added "with written permission from BMR" to line 111
- We added "Both BMR collection sites experience marine rather than estuarine conditions; mean daily salinity at BMR was $33.4 \pm 0.34$ (1 SD) PSU in 2018." (Lines 118-119)
- We added that the Olympus BH–2 light microscope was "equipped with both transmitted and reflected light sources" (line 146).
- We added "In some cases, it was difficult to visually determine the terminal band color; to supplement visual inspection, gray values were obtained from the 8–bit image through a transect of dark-light banding at the region of interest (Katayama and Isshiki, 2007; Fig. 2b). Using the transect tool and the "Plot Profile" command in Fiji, we obtained a grayscale profile and gray values for each transect of all 40 specimen images. To determine the proportion of light banding in each individual specimen and confirm the coloration of the terminal band, gray values greater than each individual's mean gray value were considered light bands, and gray values less than each individual's mean were considered dark bands." (Lines 152-158)
- We added "Percent of light bands was calculated as (total light band amount (mm)/(total dark + light band amount (mm) x 100%. Microsoft Excel was used to

calculate and standardize each specimen's gray–value variance as a proxy for band expression; a higher standardized gray–value variance indicated greater contrast (strong visual expression) between dark–light bands, while low standardized gray–value variance indicated low contrast (weak visual expression) between dark–light bands. We expected greater contrast (higher gray-value variance) to correspond to more "normal" growth patterns (i.e., alternating deposition of distinguishable dark and light layers) and lower contrast (lower gray-value variance) to correspond to more disturbances or intervals of halted growth (i.e., more dark banding or little difference between dark and light bands)." (Lines 158-164)

- We updated Figure 6 on page 15 (specifically, we changed Figure 6b to feature seasonal SST *range* rather than cumulated average seasonal SST on the x-axis. This figure now shows the relationship between season of collection and terminal band color more effectively.)
- We added the Dodd (1963) citation to support our description of *M. californianus* shell mineralogy (line 380) with the use of X-Ray Diffraction (XRD)
- We added the Gordon and Carriker (1978) citation (line 405), Schöne and Surge (2012) citation (line 408), and the following explanation: "While it has been suggested that *M. californianus* partially dissolves its own shell during anaerobiosis in order to neutralize the acidic end–products of its own metabolism (Gordon and Carriker, 1978; McCoy et al., 2011), other sclerochronological studies have dismissed the anaerobiosis-dissolution theory to argue instead that the growth band pattern in bivalve molluscs is the visual result of fluctuating calcification rates and changes in crystallographic size and orientation (Schöne and Surge, 2012). While the mechanism that produces dark bands in *M. californianus* has not been identified, dark bands could be (1) dissolution bands, (2) organic–rich bands, (3) the visual expression of slow calcite biomineralization during sub–optimal growing conditions, or (4) a combination of multiple processes. Further investigation of the relationships (if any) among anaerobiosis, dissolution, and growth features in intertidal bivalves like *M. californianus* would help elucidate the mechanism of growth band formation in greater detail." (Lines 404-412)
- We cited the Bullard et al. (2021) paper again in line 532
- We added that "the primary challenge with sampling the inner calcite layer for stable isotope analysis is the fine–resolution sampling required; the inner calcite layer is ~ 2 mm thick (mean of n = 40) and individual bands can be extremely thin (on the order of micrometers)." (Lines 539-541)
- We added "While the three collection sites experience synchronous warm or cool periods, it is possible that small-scale oceanographic variability results in different shell growth patterns at the BMR versus Portuguese Beach open-coast collection sites. However, we found evidence to suggest that relative (rather than absolute) temperature variability is a stronger influence over shell growth patterns (Figs. 6, 7)." (Lines 549-552)

**Response to RC1**

RC1: Referee #1 comment (in gray)
AC: Author comment (in black)

AC: We thank Referee #1 (Daniel Killam) for the supportive feedback and helpful comments to improve this manuscript. We have addressed each comment below and will incorporate the suggestions in the manuscript.

RC1: The study from Vriesman et al is an investigation into biomineralization patterns of *Mytilus californianus*, reporting on the potential biological and environmental causes of the semi-periodic growth lines present in the shells of this species. *M. californianus* is an iconic and well-studied species in the fields of intertidal ecology and marine invertebrate physiology, but is comparatively under-studied in terms of its biomineralization and sclerochronology. This is partially because as the authors note, mytilids are enigmatic in their shell growth patterns, often lacking the clear, consistent annual increments of shells in other bivalve taxa. Thus, the work of Vriesman et al represents a long-overdue sclerochronological revisitation of *M. californianus*. The study provides a characterization of the tripartite shell layer structure of *M. californianus*, which is unusual among the mytilids and a point of error in some recent studies of the species (who shall remain unnamed), and then investigates environmental determinants of growth bands within each respective shell layer. Prior sclerochronological work on this species has been stymied by the lack of true periodic growth bands, preventing the creation of an age model, so the authors take the alternative approach of characterizing the terminal growth band (dark or light) and the environmental conditions concurrent with those bands. They propose that the formation of light bands is often concurrent with "goldilocks" (my term) conditions associated with stable, moderate temperatures and a lack of upwelling. The study also looks at whether variations in the contrast of the dark-light bands might have environmental significance related to microenvironment and other factors. As such, the study represents a worthwhile addition to the limited literature on mytilid sclerochronology and I recommend its publication. Below I provide line-by-line questions/comments/suggestions that came to mind while reading.

AC: We greatly appreciate the feedback regarding our manuscript's contribution to sclerochronology and we value the line-by-line questions/comments/suggestions (addressed below). We will incorporate them into the manuscript as well.

RC1: 105: If permission or permit from the reserve was required for collection, mention that here.

AC: We obtained a permit to access and collect *M. californianus* shells directly from the Bodega Marine Reserve (BMR) since it is part of the UC Natural Reserve System. We will specify this in the Methods.

RC1: 138: Were you able to identify the terminal band as dark or light easily across all shells? Or were there edge cases where identification was difficult, such as for the shells with low contrast? Your Fig. S3 was helpful as an example.

AC: In some cases, it was difficult to definitely identify the terminal band as either light or dark with visual inspection alone. We will state this explicitly in section 2.3, and we will update our methodology to be transparent about how we identified band coloration in these cases. Lines ~160-164 will be edited to the following: "In some cases, it was difficult to visually determine the terminal band color; to supplement visual inspection, gray values were obtained from the 8–bit image through a transect of dark-light banding at the region of interest. To determine the proportion of light banding in each individual specimen and confirm the coloration of the terminal band, gray values greater than the mean were considered light bands, and gray values less than the mean were considered dark bands."

RC1: 143: Reflected, transmitted light or both?

AC: We used a microscope equipped with both transmitted and reflected light sources. We used both for our analysis but the photographs shown in the paper were taken under reflected light. We will specify that in the Methods section. We will also add this to the Figure 2 caption.

RC1: 153: This gray-value variance approach seems to me rather novel and merits greater elaboration in the methods. Have any other references used a similar approach? I couldn't find too many prior uses of this technique; one for fish otoliths (https://doi.org/10.1016/j.seares.2006.09.006) but not a whole lot else. Did you have any prior expectation of what these results would mean? I.e. did you expect greater contrast to correspond to greater growth disruption? Also, for reproducibility, provide more info on how you collected and standardized the gray values. Was this via the transect tool in Fiji/ImageJ?

AC: We thank the reviewer for the interest in our gray-value variance technique. We developed this technique for this paper in particular and we are not aware of any other references that utilize gray-value variance as an estimate of dark-light band expression. We were aware of the Katayama and Isshiki (2007) paper, which uses image opacity and gray values to examine otolith structure. While we did not base our methodology off of this, this is a valuable reference to explain the use of imaging software to examine growth structures, so will cite this paper in our methods (section 2.3).

We developed they gray-value variance method after we had made first-order observations of all 40 thin sections; we noticed that many samples had strongly expressed, visually clear growth bands (i.e., ideal for a sclerochronologist). Other samples had weakly expressed, cloudy bands that made it more difficult to distinguish dark from light (i.e., very poor for sclerochronological analysis). We were curious if the variation in growth band expression/clarity was due to micro-environment and/or a temporal shift. In this case, we expected greater contrast (higher gray-value variance) to correspond to more "normal" growth patterns (i.e., alternating deposition of distinguishable dark and light layers) and lower contrast (lower gray-value variance) to correspond to more disturbances or intervals of halted growth (i.e., more dark banding or little difference between dark and light bands).

We will specify that this technique was carried out using the transect tool in Fiji. We will also elaborate further on how we obtained gray values, how we calculated and standardized gray-value variance, and how we interpreted gray-value variance.

RC1: 163: I assume the percent of light bands was calculated as (light band number)/(total dark + light band number)*100%? Might want to note that explicitly.

AC: Yes, this is correct. We will add this immediately after our explanation of how gray values were used to determine band color.

RC1: 202: Can you provide more background on your identifications of polymorphs for each respective layer? Is this based on the prior observations of mineralogy of this species, or were you also identifying based on their microstructural appearance, response to plane polarized light, etc?

AC: We appreciate the interest in the calcium carbonate polymorphs in the shell of *M. californianus*. Referee #2 asked a similar question and recommended using X-ray diffraction (XRD) to determine the mineralogy of this species. We value both questions and we have future high-resolution imaging analysis planned on this species to observe crystallographic orientation, micro-fabrics, and crystal sizes on a much finer scale. For the present study, we were able to visually distinguish calcite from aragonite based on their appearances under reflected light microscope. The inner and outer calcite layers have a blade-like prismatic microstructure. Mutvei's solution accentuated the appearance of these blade-like calcite prisms, which lay perpendicular to the shell exterior. The middle aragonite layer has a brick-like microstructure, with aragonitic 'bricks' laying parallel to the shell's exterior. Extensive visual observation of many *M. californianus* thin sections gave us confidence in our identification of three mineralogical layers. We will also cite Dodd (1963) in addition to Dodd (1964), which both utilized XRD to determine the mineralogy of *M. californianus*.

RC1: 228: Do you have any data on the average thickness of these different types of bands? A quick mention of those descriptive stats would assist in placing these bands in context relative to the animal's shell height.

AC: This is a very good question. While we were collecting shell characteristic data, we intended to measure the thickness of each dark and light band in every specimen. We discovered that this would be difficult since dark-light bands often taper, appear at an angle, and/or are inconsistent throughout the shell. For this reason, we chose to measure/describe characteristics that we could more definitively quantify or distinguish (e.g., terminal band color, thickness of the inner calcite layer at the region of interest, growth band expression, etc.). Figure 6 contains information about the widths of the terminal bands, which are ~ 0.15 to ~ 0.8 mm thick and Table S1 contains cross-sectional thickness of the inner calcite layer (0.3 mm to 3.6 mm thick). The thicknesses of dark and light bands are highly variable, but all are on the order of a few hundred micrometers or less. This makes fine-resolution sub-sampling (such as drilling for oxygen isotope analysis) extremely difficult in this inner calcite region. This point relates to a question posed by Referee #2, so we will address the narrowness of this region and the fine scale of the banding in the Discussion.

RC1: 380: You could note here that the anaerobosis-dissolution idea had been originally proposed as a mechanism for growth line formation across bivalves including subtidal taxa like Mercenaria, but has been since been dismissed by some workers (see Schone and Surge,

sclerochronology treatise chapter). However, it is still a theory of interest in intertidal taxa like mytilids, which some work (including the McCoy study you cite) has determined have much greater swings in intra-shell redox conditions during tidal emersion. Basically, bringing up the anaerobosis theory could be controversial in some quarters of the sclerochronological community (I would not be surprised if the other reviewer has reservations), but it seems your results in this paper merit greater investigation of whether dissolution is at play in the creation of dark growth bands in *M. californianus*. So you could add greater mention of the fact that the theory is still controversial but merits further investigation, and might help explain why mytilids have such unusual growth patterns compared to other bivalves.

AC: We thank the reviewer for the insights about the anaerobiosis-dissolution theory. We find this extremely interesting as well. We will add the Schöne and Surge (2012) citation to be transparent about the controversy regarding dark-light band formation in bivalves. We will also mention their alternative hypothesis (dark bands = visual expression of slower growth and smaller crystals) in the Discussion (section 4.1). We will make it clear that there are multiple theories regarding growth band formation and that further research is required to determine the mechanisms responsible for such complex growth features in intertidal bivalves like *M. californianus*. We will also add the Gordon and Carriker (1978) reference to this paragraph since this is the study cited (and rejected) by Schöne and Surge (2012).

RC1: 430: Do you have any data on emersion time at the different intertidal positions at the study sites? If so, does mean emersion time have an influence as an ordinal predictor on band contrast across sites? I just notice your MIP population has a higher variance than the other two and wonder if it's hiding a couple of subgroups. Even if you don't have data on tidal emersion time, might be useful to have the point shape in Figure 5B correspond to site of origin, to see if there's any separation.

AC: We appreciate this question; information on emersion time at each intertidal position would be extremely useful. We will find a way to achieve this for the next intertidal field experiment. To address the point about highlighting site of origin, Figure S5 features the same x- and y-axes (tidal position and standardized gray-value variance, respectively) and omits all Portuguese Beach specimens to emphasize only BMR shells. Per the reviewer's suggestion, we attempted to map point shape to site of origin in the existing Figure 5B, but this substantially changes the figure by grouping each intertidal position into separate, individual box plots (i.e. LIP would feature two boxes, MIP would feature three boxes, HIP would feature two boxes). We were concerned that this would distract from the categories of interest (intertidal position and habitat), so we concluded that leaving the figure as is and pointing the reader to Figure S5 is the best way to address the reviewer's question.

RC1: 510: You could cite the Bullard study again here.

AC: We will cite the Bullard et al. (2021) paper again here and mention that it documented a recent decline in aragonite relative to calcite in *M. californianus* shells from southern California. It fits in very well here and we thank the reviewer for this suggestion.

**Response to RC2**

RC2: Referee #2 comment (in gray)
AC: Author comment (in black)

AC: We thank Referee #2 (A.D. Wanamaker Jr.) for the insightful suggestions to help improve this manuscript. We have addressed each suggestion below and will incorporate them into the manuscript.

RC2: The manuscript is clearly written and the results suggest that the banding pattern (light and dark couplets) in Mytilus californianus is largely associated with environmental conditions. A real strength of the study is the abundant environmental data from which the shells were collected. This allowed the authors to investigate which parameters might be most important in controlling the light and dark banding in the shell.

I mostly have some small suggestions that will hopefully make your statements/conclusions a bit stronger and a few editorial suggestions that might improve the flow of the manuscript. Overall, I think this is a strong contribution to the field of sclerochronology.

AC: We greatly appreciate this feedback and value the suggestions to improve the manuscript. We will address each comment, below.

RC2: If you provided additional evidence from x-ray diffraction (XRD) that you have three distinct mineral layers, that would be stronger than the optically derived evidence. Because this is a major finding of this study, this additional line of evidence is warranted. Furthermore, this will be the "go to paper" to cite this mineralogical finding. XRD is quick and relatively inexpensive.

AC: We appreciate the interest in the mineralogical layering of *M. californianus* and we agree that further XRD analysis would be a valuable contribution to the *M. californianus* literature. We are currently planning a study that will feature high-resolution imaging on this species, including XRD, scanning electron microscopy (SEM), and electron backscatter diffraction (EBSD) to examine micro-textures, crystallographic orientation, and crystal sizes. The goal of the present study was to perform optical analysis of *M. californianus* shells to visually characterize the shell structure and determine relationships between environmental conditions and growth band pattern. We were able to identify three distinct layers: (1) in whole valves, where a white chalky calcite layer spanning from umbo to the midpoint of the shell's interior can be observed visually and tactilely, (2) under reflected light microscope, where the microstructural appearance of calcite crystal fabric is distinct from aragonite crystal fabric, and (3) under transmitted light microscope after etching with Mutvei's solution, which accentuates the shape of prismatic calcite crystals. In thin section, the inner and outer calcite layers have a blade-like prismatic microstructure. After immersing in Mutvei's solution, the blade-like calcite prisms laying perpendicular to the shell exterior are even more pronounced. The middle aragonite layer has a brick-like microstructure, with aragonitic 'bricks' laying parallel to the shell's exterior. The aragonite layer also characteristically forms a 'zig-zag' pattern (Dodd, 1964) that we identified in

our samples as well. With the reviewer's comment in mind, we revisited Dodd (1964) (which cites Dodd (1963)) and found that both papers performed XRD analysis to determine mineralogy in *M. californianus*. We will add the Dodd (1963) reference to the Discussion (section 4.1) and make it clear that Dodd (1963; 1964) determined the mineralogy of this species using XRD analysis, and that we were able to confirm this visually using optical microscopy. While some recent studies have mischaracterized the shell structure of *M. californianus*, we suspect that this is because this species is commonly assumed to have only an inner aragonite layer like the rest of its congeners and that its inner calcite layer is therefore frequently overlooked.

RC2: Do you have modern shells from Portuguese Beach? If not, you are "making the argument" that site 3 and site 2 (open coast environments) are similar enough to suggest that changes in shell growth between the modern and archival specimens is related to time dependent growth changes rather than a difference in growth from two different locations (i.e., growth is different because they are at different sites). I think it is warranted to add something to the discussion about this assumption.

AC: We thank the reviewer for the interest in the Portuguese Beach mussels. These were collected by Michael Kennedy in 2002 and 2003 for his dissertation work and donated to us in 2019 by one of his dissertation committee members (Ann Russell). Unfortunately, we had no access to modern shells from Portuguese Beach during the period that we were analyzing shell characteristics for this study; the California Dept. of Fish and Wildlife permitting/licensing centers were closed during this point of the COVID-19 pandemic. We consulted Bodega Marine Laboratory oceanographer Dr. John Largier to ensure that BMR and Portuguese Beach would be similar enough to draw conclusions about time-dependent growth trends. According to J. Largier (pers. comm., 2021), there is little oceanographic difference associated with the 7 km of alongshore separation between site 2 and site 3. Sonoma Coast is well studied oceanographically; it is all part of the same upwelling cell (we cite Largier et al., 1993 to support this), so sites all along this stretch of coast are all well correlated with cold (or warm) periods occurring synchronously at all sites up and down the coast. While we address this in the Methods (section 2.1), we will mention this in the Discussion (section 4.4) as a potential source of variability between modern and archival growth patterns.

RC2: Line – 39 – after ~ 500 years add Butler et al., 2013; Butler, P. G., A. D. Wanamaker, J. D. Scourse, C. A. Richardson, and D. J. Reynolds (2013), Variability of marine climate on the North Icelandic Shelf in a 1357-year proxy archive based on growth increments in the bivalve Arctica islandica, Palaeogeography Palaeoclimatology Palaeoecology, 373, 141-151, doi: 10.1016/j.palaeo.2012.01.016

AC: We agree with the suggestion to cite the Butler et al. (2013) paper. We will incorporate this reference here.

RC2: not everyone would support this statement about obvious/clear daily growth increments in A. islandica. Better to say Schone et al concluded …

AC: We agree and will rephrase this sentence by adding in the word "concluded" to reflect that we are reporting Schone et al. (2005a) and Schone (2013)'s interpretations regarding daily growth increments.

RC2: Table 1 – consider adding Wanamaker et al 2008 for Mg/Ca in Mytilus edulis
Wanamaker, A. D., K. J. Kreutz, T. Wilson, H. W. Borns, D. S. Introne, and S. Feindel (2008), Experimentally determined Mg/Ca and Sr/Ca ratios in juvenile bivalve calcite for Mytilus edulis: implications for paleotemperature reconstructions, Geo-Marine Letters, 28(5-6), 359-368.
We found differing Mg/Ca ratio relationships based on ambient seawater salinity. Thus, there is likely a physiological response/control over elemental incorporation.

AC: We value the reference suggestion and we will add the Wanamaker et al. (2008) paper to the Mg/Ca row of Table 1.

RC2: I think the last paragraph in the Introduction should be the aims of the study. Thus, I suggest making the paragraph (line 75) about banding the first paragraph of the Introduction. I think the Introduction lost clarity after reading about the aims which was followed by a very broad discussion of banding.

AC: We thank the reviewer for this suggestion. We agree that this paragraph about banding feels somewhat misplaced; it was moved around multiple times during the initial writing of this manuscript. We will move it back up to the first paragraph so that the Introduction (section 1.1) ends with the enumerated study objectives.

RC2: Line 116 – add standard deviation to salinity range and report if it is 1 or 2 standard deviations.

AC: In 2018, mean daily BOON salinity was 33.4 PSU with 1 standard deviation (SD) of 0.34. We will report this here. We used the year 2018 to calculate mean daily salinity since the BOON instrument malfunctioned in 2019 and 2020; 2018 was the most complete record for salinity.

RC2: I found figure 6 a bit confusing/hard to follow. Perhaps adding "range" to Seasonal SST for the x-axis on panel B would eliminate the possibility of thinking panels A and b are nearly identical.

AC: We thank the reviewer for this suggestion, which we agree would greatly improve the figure. We will re-work Figure 6b to feature seasonal range on the x-axis and change the point shape to denote the season of collection to highlight the relationship between season, seasonal temperature range, and terminal band color. See the last page of this PDF document for the revised figure and new caption.

RC2: Line 390- onward – high resolution sampling (representing weekly or so) of oxygen isotopes in the outer calcite layer would help solve this issue right? Is this planned? Some discussion of this possibility is warranted in the Discussion (or future work?).

AC: We agree that stable isotope analysis would be useful for estimating growth rate and reconstructing annual cycles in *M. californianus* shells. Multiple studies cited in our paper (e.g., Jones and Kennett (1999), Jazwa and Kennett (2016), Ford et al. (2010)) performed high-resolution (weekly to monthly) subsampling of the outer calcite layer of *M. californianus* shells and documented seasonal oscillations of oxygen isotopes recorded by this species. While we could perform the same analyses (high-resolution oxygen isotope sampling of the outer calcite layer) in our samples, it would be difficult to compare this outer calcite oxygen isotope profile to dark-light banding in the inner calcite layer in any accurate or useful way. We know that the outer calcite layer grows extensionally and the inner calcite layer grows inward adding to shell thickness, but we do not know if these layers calcify simultaneously or at proportional rates. The banding in each layer is certainly a different resolution (i.e., tidal banding in the outer calcite layer and ~ seasonal to multi-annual banding in the inner calcite layer based on environmental conditions). Ideally, one could sample both the inner and outer calcite layer at a very high resolution and then match up the two sinusoidal oxygen isotope profiles to (1) estimate and compare growth rates of each layer and (2) estimate the amount of time represented by a pair of dark-light bands. Unfortunately, this would require *in situ* oxygen isotope analysis since drilling/milling for powdered samples would not be possible in the inner calcite layer, which is very thin (~ 2 mm on average) and could not accommodate a drill bit. Further, dark-light bands are extremely thin (on the order of microns), which would be challenging to sample even with costly instrumentation such as secondary-ion mass spectrometry (SIMS) or sensitive high-resolution ion microprobe (SHRIMP). We will address this challenge briefly in the Discussion (section 4.3).

RC2: Also, when considering future work, Goodwin et al found that Mercenaria mercenaria clams grew during the warmest part of the day throughout the year whereas oysters in the same setting had no preference. If we were to sample these clams and oysters for oxygen isotopes, we might then conclude that they grew in different environments, but they did not. Thus, I wonder if monitoring daily high and low temperatures might provide some additional insight on your work. This is just a thought- no action needed.
Goodwin, D.H., Gillikin, D.P., Jorn, E.N., Fratian, M.C., and Wanamaker, A.D., (2021) Comparing contemporary biological archives from Mercenaria mercenaria and Crassostrea virginica: Insights on paleoenvironmental reconstructions, Palaeogeography, Palaeclimatology, Palaeoecology, 562, doi.org/10.1016/j.palaeo.2020.110110.

AC: We thank Referee #2 for this thoughtful comment. We will read the Goodwin et al. (2021) paper recommended here and consider incorporating tidal SST monitoring for future work.

[Figure]

**Figure 6.** Relationships between SST, width of terminal band, color of terminal band, specimen type, and collection season for all 40 specimens. Filled (black) points represent dark terminal bands and open (white) points represent light terminal bands in both plots. (a) Relationships between mean monthly SST during month of collection, terminal band color, and terminal band width. Shaded bar represents approximate monthly SST range over which most specimens are associated with light terminal bands (12.75-13.5°C). Specimen type specified with point shapes (see legend A). (b) Relationships between the seasonal SST range (mean daily SST maximum – mean daily SST minimum for each season of collection), terminal band color, and terminal band width. Shaded bar highlights that most terminal light bands are associated with a seasonal SST range < 5°C. Collection season specified with point shapes (see legend B).

**Response to RC1**

RC1: Referee #1 comment (in gray)
AC: Author comment (in black)

AC: We thank Referee #1 (Daniel Killam) for the supportive feedback and helpful comments to improve this manuscript. We have addressed each comment below and will incorporate the suggestions in the manuscript.

RC1: The study from Vriesman et al is an investigation into biomineralization patterns of *Mytilus californianus*, reporting on the potential biological and environmental causes of the semi-periodic growth lines present in the shells of this species. *M. californianus* is an iconic and well-studied species in the fields of intertidal ecology and marine invertebrate physiology, but is comparatively under-studied in terms of its biomineralization and sclerochronology. This is partially because as the authors note, mytilids are enigmatic in their shell growth patterns, often lacking the clear, consistent annual increments of shells in other bivalve taxa. Thus, the work of Vriesman et al represents a long-overdue sclerochronological revisitation of *M. californianus*. The study provides a characterization of the tripartite shell layer structure of *M. californianus*, which is unusual among the mytilids and a point of error in some recent studies of the species (who shall remain unnamed), and then investigates environmental determinants of growth bands within each respective shell layer. Prior sclerochronological work on this species has been stymied by the lack of true periodic growth bands, preventing the creation of an age model, so the authors take the alternative approach of characterizing the terminal growth band (dark or light) and the environmental conditions concurrent with those bands. They propose that the formation of light bands is often concurrent with "goldilocks" (my term) conditions associated with stable, moderate temperatures and a lack of upwelling. The study also looks at whether variations in the contrast of the dark-light bands might have environmental significance related to microenvironment and other factors. As such, the study represents a worthwhile addition to the limited literature on mytilid sclerochronology and I recommend its publication. Below I provide line-by-line questions/comments/suggestions that came to mind while reading.

AC: We greatly appreciate the feedback regarding our manuscript's contribution to sclerochronology and we value the line-by-line questions/comments/suggestions (addressed below). We will incorporate them into the manuscript as well.

RC1: 105: If permission or permit from the reserve was required for collection, mention that here.

AC: We obtained a permit to access and collect *M. californianus* shells directly from the Bodega Marine Reserve (BMR) since it is part of the UC Natural Reserve System. We will specify this in the Methods.

RC1: 138: Were you able to identify the terminal band as dark or light easily across all shells? Or were there edge cases where identification was difficult, such as for the shells with low contrast? Your Fig. S3 was helpful as an example.

AC: In some cases, it was difficult to definitely identify the terminal band as either light or dark with visual inspection alone. We will state this explicitly in section 2.3, and we will update our methodology to be transparent about how we identified band coloration in these cases. Lines ~160-164 will be edited to the following: "In some cases, it was difficult to visually determine the terminal band color; to supplement visual inspection, gray values were obtained from the 8–bit image through a transect of dark-light banding at the region of interest. To determine the proportion of light banding in each individual specimen and confirm the coloration of the terminal band, gray values greater than the mean were considered light bands, and gray values less than the mean were considered dark bands."

RC1: 143: Reflected, transmitted light or both?

AC: We used a microscope equipped with both transmitted and reflected light sources. We used both for our analysis but the photographs shown in the paper were taken under reflected light. We will specify that in the Methods section. We will also add this to the Figure 2 caption.

RC1: 153: This gray-value variance approach seems to me rather novel and merits greater elaboration in the methods. Have any other references used a similar approach? I couldn't find too many prior uses of this technique; one for fish otoliths (https://doi.org/10.1016/j.seares.2006.09.006) but not a whole lot else. Did you have any prior expectation of what these results would mean? I.e. did you expect greater contrast to correspond to greater growth disruption? Also, for reproducibility, provide more info on how you collected and standardized the gray values. Was this via the transect tool in Fiji/ImageJ?

AC: We thank the reviewer for the interest in our gray-value variance technique. We developed this technique for this paper in particular and we are not aware of any other references that utilize gray-value variance as an estimate of dark-light band expression. We were aware of the Katayama and Isshiki (2007) paper, which uses image opacity and gray values to examine otolith structure. While we did not base our methodology off of this, this is a valuable reference to explain the use of imaging software to examine growth structures, so will cite this paper in our methods (section 2.3).

We developed they gray-value variance method after we had made first-order observations of all 40 thin sections; we noticed that many samples had strongly expressed, visually clear growth bands (i.e., ideal for a sclerochronologist). Other samples had weakly expressed, cloudy bands that made it more difficult to distinguish dark from light (i.e., very poor for sclerochronological analysis). We were curious if the variation in growth band expression/clarity was due to micro-environment and/or a temporal shift. In this case, we expected greater contrast (higher gray-value variance) to correspond to more "normal" growth patterns (i.e., alternating deposition of distinguishable dark and light layers) and lower contrast (lower gray-value variance) to correspond to more disturbances or intervals of halted growth (i.e., more dark banding or little difference between dark and light bands).

We will specify that this technique was carried out using the transect tool in Fiji. We will also elaborate further on how we obtained gray values, how we calculated and standardized gray-value variance, and how we interpreted gray-value variance.

RC1: 163: I assume the percent of light bands was calculated as (light band number)/(total dark + light band number)*100%? Might want to note that explicitly.

AC: Yes, this is correct. We will add this immediately after our explanation of how gray values were used to determine band color.

RC1: 202: Can you provide more background on your identifications of polymorphs for each respective layer? Is this based on the prior observations of mineralogy of this species, or were you also identifying based on their microstructural appearance, response to plane polarized light, etc?

AC: We appreciate the interest in the calcium carbonate polymorphs in the shell of *M. californianus*. Referee #2 asked a similar question and recommended using X-ray diffraction (XRD) to determine the mineralogy of this species. We value both questions and we have future high-resolution imaging analysis planned on this species to observe crystallographic orientation, micro-fabrics, and crystal sizes on a much finer scale. For the present study, we were able to visually distinguish calcite from aragonite based on their appearances under reflected light microscope. The inner and outer calcite layers have a blade-like prismatic microstructure. Mutvei's solution accentuated the appearance of these blade-like calcite prisms, which lay perpendicular to the shell exterior. The middle aragonite layer has a brick-like microstructure, with aragonitic 'bricks' laying parallel to the shell's exterior. Extensive visual observation of many *M. californianus* thin sections gave us confidence in our identification of three mineralogical layers. We will also cite Dodd (1963) in addition to Dodd (1964), which both utilized XRD to determine the mineralogy of *M. californianus*.

RC1: 228: Do you have any data on the average thickness of these different types of bands? A quick mention of those descriptive stats would assist in placing these bands in context relative to the animal's shell height.

AC: This is a very good question. While we were collecting shell characteristic data, we intended to measure the thickness of each dark and light band in every specimen. We discovered that this would be difficult since dark-light bands often taper, appear at an angle, and/or are inconsistent throughout the shell. For this reason, we chose to measure/describe characteristics that we could more definitively quantify or distinguish (e.g., terminal band color, thickness of the inner calcite layer at the region of interest, growth band expression, etc.). Figure 6 contains information about the widths of the terminal bands, which are ~ 0.15 to ~ 0.8 mm thick and Table S1 contains cross-sectional thickness of the inner calcite layer (0.3 mm to 3.6 mm thick). The thicknesses of dark and light bands are highly variable, but all are on the order of a few hundred micrometers or less. This makes fine-resolution sub-sampling (such as drilling for oxygen isotope analysis) extremely difficult in this inner calcite region. This point relates to a question posed by Referee #2, so we will address the narrowness of this region and the fine scale of the banding in the Discussion.

RC1: 380: You could note here that the anaerobosis-dissolution idea had been originally proposed as a mechanism for growth line formation across bivalves including subtidal taxa like Mercenaria, but has been since been dismissed by some workers (see Schone and Surge,

sclerochronology treatise chapter). However, it is still a theory of interest in intertidal taxa like mytilids, which some work (including the McCoy study you cite) has determined have much greater swings in intra-shell redox conditions during tidal emersion. Basically, bringing up the anaerobosis theory could be controversial in some quarters of the sclerochronological community (I would not be surprised if the other reviewer has reservations), but it seems your results in this paper merit greater investigation of whether dissolution is at play in the creation of dark growth bands in *M. californianus*. So you could add greater mention of the fact that the theory is still controversial but merits further investigation, and might help explain why mytilids have such unusual growth patterns compared to other bivalves.

AC: We thank the reviewer for the insights about the anaerobiosis-dissolution theory. We find this extremely interesting as well. We will add the Schöne and Surge (2012) citation to be transparent about the controversy regarding dark-light band formation in bivalves. We will also mention their alternative hypothesis (dark bands = visual expression of slower growth and smaller crystals) in the Discussion (section 4.1). We will make it clear that there are multiple theories regarding growth band formation and that further research is required to determine the mechanisms responsible for such complex growth features in intertidal bivalves like *M. californianus*. We will also add the Gordon and Carriker (1978) reference to this paragraph since this is the study cited (and rejected) by Schöne and Surge (2012).

RC1: 430: Do you have any data on emersion time at the different intertidal positions at the study sites? If so, does mean emersion time have an influence as an ordinal predictor on band contrast across sites? I just notice your MIP population has a higher variance than the other two and wonder if it's hiding a couple of subgroups. Even if you don't have data on tidal emersion time, might be useful to have the point shape in Figure 5B correspond to site of origin, to see if there's any separation.

AC: We appreciate this question; information on emersion time at each intertidal position would be extremely useful. We will find a way to achieve this for the next intertidal field experiment. To address the point about highlighting site of origin, Figure S5 features the same x- and y-axes (tidal position and standardized gray-value variance, respectively) and omits all Portuguese Beach specimens to emphasize only BMR shells. Per the reviewer's suggestion, we attempted to map point shape to site of origin in the existing Figure 5B, but this substantially changes the figure by grouping each intertidal position into separate, individual box plots (i.e. LIP would feature two boxes, MIP would feature three boxes, HIP would feature two boxes). We were concerned that this would distract from the categories of interest (intertidal position and habitat), so we concluded that leaving the figure as is and pointing the reader to Figure S5 is the best way to address the reviewer's question.

RC1: 510: You could cite the Bullard study again here.

AC: We will cite the Bullard et al. (2021) paper again here and mention that it documented a recent decline in aragonite relative to calcite in *M. californianus* shells from southern California. It fits in very well here and we thank the reviewer for this suggestion.

**Response to RC2**

RC2: Referee #2 comment (in gray)
AC: Author comment (in black)

AC: We thank Referee #2 (A.D. Wanamaker Jr.) for the insightful suggestions to help improve this manuscript. We have addressed each suggestion below and will incorporate them into the manuscript.

RC2: The manuscript is clearly written and the results suggest that the banding pattern (light and dark couplets) in Mytilus californianus is largely associated with environmental conditions. A real strength of the study is the abundant environmental data from which the shells were collected. This allowed the authors to investigate which parameters might be most important in controlling the light and dark banding in the shell.

I mostly have some small suggestions that will hopefully make your statements/conclusions a bit stronger and a few editorial suggestions that might improve the flow of the manuscript. Overall, I think this is a strong contribution to the field of sclerochronology.

AC: We greatly appreciate this feedback and value the suggestions to improve the manuscript. We will address each comment, below.

RC2: If you provided additional evidence from x-ray diffraction (XRD) that you have three distinct mineral layers, that would be stronger than the optically derived evidence. Because this is a major finding of this study, this additional line of evidence is warranted. Furthermore, this will be the "go to paper" to cite this mineralogical finding. XRD is quick and relatively inexpensive.

AC: We appreciate the interest in the mineralogical layering of *M. californianus* and we agree that further XRD analysis would be a valuable contribution to the *M. californianus* literature. We are currently planning a study that will feature high-resolution imaging on this species, including XRD, scanning electron microscopy (SEM), and electron backscatter diffraction (EBSD) to examine micro-textures, crystallographic orientation, and crystal sizes. The goal of the present study was to perform optical analysis of *M. californianus* shells to visually characterize the shell structure and determine relationships between environmental conditions and growth band pattern. We were able to identify three distinct layers: (1) in whole valves, where a white chalky calcite layer spanning from umbo to the midpoint of the shell's interior can be observed visually and tactilely, (2) under reflected light microscope, where the microstructural appearance of calcite crystal fabric is distinct from aragonite crystal fabric, and (3) under transmitted light microscope after etching with Mutvei's solution, which accentuates the shape of prismatic calcite crystals. In thin section, the inner and outer calcite layers have a blade-like prismatic microstructure. After immersing in Mutvei's solution, the blade-like calcite prisms laying perpendicular to the shell exterior are even more pronounced. The middle aragonite layer has a brick-like microstructure, with aragonitic 'bricks' laying parallel to the shell's exterior. The aragonite layer also characteristically forms a 'zig-zag' pattern (Dodd, 1964) that we identified in

our samples as well. With the reviewer's comment in mind, we revisited Dodd (1964) (which cites Dodd (1963)) and found that both papers performed XRD analysis to determine mineralogy in *M. californianus*. We will add the Dodd (1963) reference to the Discussion (section 4.1) and make it clear that Dodd (1963; 1964) determined the mineralogy of this species using XRD analysis, and that we were able to confirm this visually using optical microscopy. While some recent studies have mischaracterized the shell structure of *M. californianus*, we suspect that this is because this species is commonly assumed to have only an inner aragonite layer like the rest of its congeners and that its inner calcite layer is therefore frequently overlooked.

RC2: Do you have modern shells from Portuguese Beach? If not, you are "making the argument" that site 3 and site 2 (open coast environments) are similar enough to suggest that changes in shell growth between the modern and archival specimens is related to time dependent growth changes rather than a difference in growth from two different locations (i.e., growth is different because they are at different sites). I think it is warranted to add something to the discussion about this assumption.

AC: We thank the reviewer for the interest in the Portuguese Beach mussels. These were collected by Michael Kennedy in 2002 and 2003 for his dissertation work and donated to us in 2019 by one of his dissertation committee members (Ann Russell). Unfortunately, we had no access to modern shells from Portuguese Beach during the period that we were analyzing shell characteristics for this study; the California Dept. of Fish and Wildlife permitting/licensing centers were closed during this point of the COVID-19 pandemic. We consulted Bodega Marine Laboratory oceanographer Dr. John Largier to ensure that BMR and Portuguese Beach would be similar enough to draw conclusions about time-dependent growth trends. According to J. Largier (pers. comm., 2021), there is little oceanographic difference associated with the 7 km of alongshore separation between site 2 and site 3. Sonoma Coast is well studied oceanographically; it is all part of the same upwelling cell (we cite Largier et al., 1993 to support this), so sites all along this stretch of coast are all well correlated with cold (or warm) periods occurring synchronously at all sites up and down the coast. While we address this in the Methods (section 2.1), we will mention this in the Discussion (section 4.4) as a potential source of variability between modern and archival growth patterns.

RC2: Line – 39 – after ~ 500 years add Butler et al., 2013; Butler, P. G., A. D. Wanamaker, J. D. Scourse, C. A. Richardson, and D. J. Reynolds (2013), Variability of marine climate on the North Icelandic Shelf in a 1357-year proxy archive based on growth increments in the bivalve Arctica islandica, Palaeogeography Palaeoclimatology Palaeoecology, 373, 141-151, doi: 10.1016/j.palaeo.2012.01.016

AC: We agree with the suggestion to cite the Butler et al. (2013) paper. We will incorporate this reference here.

RC2: not everyone would support this statement about obvious/clear daily growth increments in *A. islandica*. Better to say Schone et al concluded …

AC: We agree and will rephrase this sentence by adding in the word "concluded" to reflect that we are reporting Schone et al. (2005a) and Schone (2013)'s interpretations regarding daily growth increments.

RC2: Table 1 – consider adding Wanamaker et al 2008 for Mg/Ca in Mytilus edulis Wanamaker, A. D., K. J. Kreutz, T. Wilson, H. W. Borns, D. S. Introne, and S. Feindel (2008), Experimentally determined Mg/Ca and Sr/Ca ratios in juvenile bivalve calcite for Mytilus edulis: implications for paleotemperature reconstructions, Geo-Marine Letters, 28(5-6), 359-368. We found differing Mg/Ca ratio relationships based on ambient seawater salinity. Thus, there is likely a physiological response/control over elemental incorporation.

AC: We value the reference suggestion and we will add the Wanamaker et al. (2008) paper to the Mg/Ca row of Table 1.

RC2: I think the last paragraph in the Introduction should be the aims of the study. Thus, I suggest making the paragraph (line 75) about banding the first paragraph of the Introduction. I think the Introduction lost clarity after reading about the aims which was followed by a very broad discussion of banding.

AC: We thank the reviewer for this suggestion. We agree that this paragraph about banding feels somewhat misplaced; it was moved around multiple times during the initial writing of this manuscript. We will move it back up to the first paragraph so that the Introduction (section 1.1) ends with the enumerated study objectives.

RC2: Line 116 – add standard deviation to salinity range and report if it is 1 or 2 standard deviations.

AC: In 2018, mean daily BOON salinity was 33.4 PSU with 1 standard deviation (SD) of 0.34. We will report this here. We used the year 2018 to calculate mean daily salinity since the BOON instrument malfunctioned in 2019 and 2020; 2018 was the most complete record for salinity.

RC2: I found figure 6 a bit confusing/hard to follow. Perhaps adding "range" to Seasonal SST for the x-axis on panel B would eliminate the possibility of thinking panels A and b are nearly identical.

AC: We thank the reviewer for this suggestion, which we agree would greatly improve the figure. We will re-work Figure 6b to feature seasonal range on the x-axis and change the point shape to denote the season of collection to highlight the relationship between season, seasonal temperature range, and terminal band color. See the last page of this PDF document for the revised figure and new caption.

RC2: Line 390- onward – high resolution sampling (representing weekly or so) of oxygen isotopes in the outer calcite layer would help solve this issue right? Is this planned? Some discussion of this possibility is warranted in the Discussion (or future work?).

AC: We agree that stable isotope analysis would be useful for estimating growth rate and reconstructing annual cycles in *M. californianus* shells. Multiple studies cited in our paper (e.g., Jones and Kennett (1999), Jazwa and Kennett (2016), Ford et al. (2010)) performed high-resolution (weekly to monthly) subsampling of the outer calcite layer of *M. californianus* shells and documented seasonal oscillations of oxygen isotopes recorded by this species. While we could perform the same analyses (high-resolution oxygen isotope sampling of the outer calcite layer) in our samples, it would be difficult to compare this outer calcite oxygen isotope profile to dark-light banding in the inner calcite layer in any accurate or useful way. We know that the outer calcite layer grows extensionally and the inner calcite layer grows inward adding to shell thickness, but we do not know if these layers calcify simultaneously or at proportional rates. The banding in each layer is certainly a different resolution (i.e., tidal banding in the outer calcite layer and ~ seasonal to multi-annual banding in the inner calcite layer based on environmental conditions). Ideally, one could sample both the inner and outer calcite layer at a very high resolution and then match up the two sinusoidal oxygen isotope profiles to (1) estimate and compare growth rates of each layer and (2) estimate the amount of time represented by a pair of dark-light bands. Unfortunately, this would require *in situ* oxygen isotope analysis since drilling/milling for powdered samples would not be possible in the inner calcite layer, which is very thin (~ 2 mm on average) and could not accommodate a drill bit. Further, dark-light bands are extremely thin (on the order of microns), which would be challenging to sample even with costly instrumentation such as secondary-ion mass spectrometry (SIMS) or sensitive high-resolution ion microprobe (SHRIMP). We will address this challenge briefly in the Discussion (section 4.3).

RC2: Also, when considering future work, Goodwin et al found that Mercenaria mercenaria clams grew during the warmest part of the day throughout the year whereas oysters in the same setting had no preference. If we were to sample these clams and oysters for oxygen isotopes, we might then conclude that they grew in different environments, but they did not. Thus, I wonder if monitoring daily high and low temperatures might provide some additional insight on your work. This is just a thought- no action needed.
Goodwin, D.H., Gillikin, D.P., Jorn, E.N., Fratian, M.C., and Wanamaker, A.D., (2021) Comparing contemporary biological archives from Mercenaria mercenaria and Crassostrea virginica: Insights on paleoenvironmental reconstructions, Palaeogeography, Palaeclimatology, Palaeoecology, 562, doi.org/10.1016/j.palaeo.2020.110110.

AC: We thank Referee #2 for this thoughtful comment. We will read the Goodwin et al. (2021) paper recommended here and consider incorporating tidal SST monitoring for future work.

[Figure]

**Figure 6.** Relationships between SST, width of terminal band, color of terminal band, specimen type, and collection season for all 40 specimens. Filled (black) points represent dark terminal bands and open (white) points represent light terminal bands in both plots. (a) Relationships between mean monthly SST during month of collection, terminal band color, and terminal band width. Shaded bar represents approximate monthly SST range over which most specimens are associated with light terminal bands (12.75-13.5°C). Specimen type specified with point shapes (see legend A). (b) Relationships between the seasonal SST range (mean daily SST maximum – mean daily SST minimum for each season of collection), terminal band color, and terminal band width. Shaded bar highlights that most terminal light bands are associated with a seasonal SST range < 5°C. Collection season specified with point shapes (see legend B).

---

## Referee Report (RR1)

The revised manuscript from Vriesman et al satisfies all of my prior suggestions, including greater elaboration in the methods section of computational and statistical techniques. They also added descriptive statistics describing the widths of the bands as requested (important for other workers comparing their shells to this study), and a mention in the discussion of the need for greater study of the link between anaerobosis and dark band formation. Regarding XRD, since that work has been conducted previously in multiple manuscripts, it is understandable that the authors worked from that precedent in their identification of shell layer mineralogy.

For the figure 5B, the authors point to a supplemental figure which does indeed satisfy my prior curiosity about the importance of site. It seems the "middle intertidal" subgroup does have greater variance in gray value even when only one site is included. I would note for future work that if the authors are using ggplot in R, there is a "group_by" argument that can create bars with different dithered point colors or shapes grouped together, but that is immaterial and the figure is fine as is.

Since all of my suggestions have been responded to and satisfied, I endorse the publication of this article with no further changes requested.

Regards,

Daniel Killam

Postdoctoral Researcher, Biosphere 2